# RINGER: CONFORMER ENSEMBLE GENERATION OF MACROCYCLIC PEPTIDES WITH SEQUENCE-CONDITIONED INTERNAL COORDINATE DIFFUSION

## ABSTRACT

Macrocyclic peptides are an emerging therapeutic modality, yet computational approaches for accurately sampling their diverse 3D ensembles remain challenging due to their conformational diversity and geometric constraints. Here, we introduce RINGER, a diffusion-based transformer model based on redundant internal coordinates that generates three-dimensional conformational ensembles of macrocyclic peptides from their 2D representations. RINGER provides fast backbone- and side-chain sampling while respecting key structural invariances of cyclic peptides. Through extensive benchmarking and analysis against gold-standard conformer ensembles of cyclic peptides generated with metadynamics, we demonstrate how RINGER generates both high-quality and diverse geometries at a fraction of the computational cost. Our work lays the foundation for improved sampling of cyclic geometries and the development of geometric learning methods for peptides.

## 1 INTRODUCTION

Macrocyclic peptides are an important therapeutic modality in modern drug discovery that occupy a unique chemical and pharmacological space between small and large molecules (Driggers et al., 2008; Muttenthaler et al., 2021; Vinogradov et al., 2019). These cyclic peptides exhibit improved structural rigidity and metabolic stability compared to their linear counterparts (Craik et al., 2013), yet retain key conformational flexibility and diversity to bind shallow protein interfaces (Villar et al., 2014). However, computational approaches for modeling their structural ensembles remain limited compared to small molecules and proteins in terms of computational speed, accuracy (sample quality), and conformational diversity (Poongavanam et al., 2018). Critically, scalable and accurate tools are necessary to enable rational design of macrocyclic drugs; access to these tools can significantly impact optimization of key properties including binding affinity (Alogheli et al., 2017; Garcia Jimenez et al., 2023), permeability (Leung et al., 2016; Rezai et al., 2006; Bhardwaj et al., 2022), and oral bioavailability (Nielsen et al., 2017).

Several key challenges hinder fast and effective macrocycle conformer generation: 1) Macrocyclic peptides exhibit diverse molecular structures and chemical modifications, including varying ring size, stereochemistry, $N$-methylation, and more (Kamenik et al., 2018). Their structural diversity, along with the increased number of rotatable bonds, results in a vast conformational space that is considerably more expensive to sample computationally. 2) Macrocycles are subject to complex non-linear constraints due to ring closure. The atomic positions, angles, and dihedrals of the macrocycle backbone are highly interdependent, and additional complex intramolecular interactions make this process inherently difficult to model (Watts et al., 2014). 3) Experimental X-ray and NMR structures for macrocycles are lacking ($\sim 10^3$) in comparison to small molecules ($\sim 10^6$ in the Cambridge Structural Database (Groom et al., 2016)) and proteins ($\sim 10^5$ in the Protein Data Bank (Berman et al., 2000)). The scarcity of available experimental data has made it difficult to integrate observational data to improve structural predictions or train machine learning-based approaches. Together, the vast conformational space combined with limited data make modeling and sampling of macrocycles not only conceptually challenging, but technically challenging due to computational cost. Existing approaches cannot accurately generate diverse and accurate conformational ensembles at a scale necessary to accelerate and enable macrocycle design.

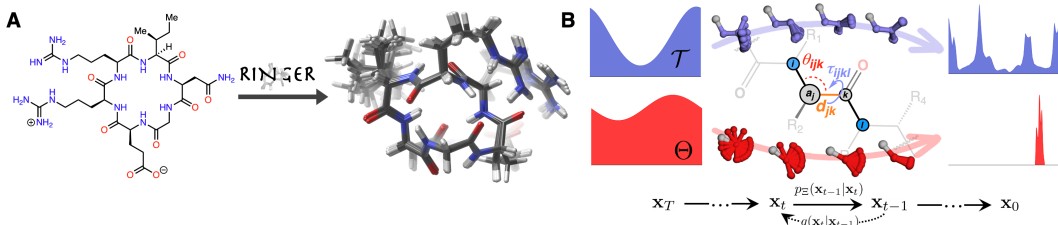

Figure 1: Overview of RINGER for macrocycle conformer generation. **A.** Given a 2D representation of a macrocyclic peptide, RINGER generates an accurate and diverse 3D conformational ensemble. **B.** An illustration of the diffusion process learning to recover the time $t = 0$ bond angle (red) and torsional (blue) distributions from time, $t = T$.

To address these limitations, we introduce RINGER (RINGER Generates Ensembles of Rings), a deep learning model designed specifically for sequence-conditioned macrocycle conformer generation (Figure 1) that efficiently samples realistic angles and torsions (i.e., internal coordinates) for macrocyclic peptides. RINGER merges a transformer architecture that naturally captures the physical equivariances and invariances of macrocyclic peptides with a discrete-time diffusion model to learn highly-coupled distributions over internal coordinates. We demonstrate how RINGER simultaneously achieves excellent performance in sample quality over angular and torsional profiles while maintaining excellent RMSDs relative to gold-standard conformer ensembles generated with the Conformer-Rotamer Ensemble Sampling Tool (CREST) (Pracht et al., 2020).

We summarize our contributions as follows:

- We propose a new framework, RINGER for all-atom conformer generation of macrocycles based on efficiently encoding their geometry using redundant internal coordinates. Our model naturally handles the cyclic nature of macrocycles and chiral side chains with both L- and D-amino acids, and we propose a simple solution to satisfy ring-closure constraints.

- We benchmark RINGER extensively against state-of-the-art physics- and machine learning-based algorithms to demonstrate how our approach better captures complex and highly-coupled geometric distributions. RINGER outperforms state-of-the-art algorithms that predict 3D conformer ensembles from their 2D structures for macrocyclic peptides.

- Our work highlights the advantages of a machine learning-based approach for conformer generation in quality, diversity, and speed. We show how RINGER generates diverse and complete conformational ensembles with excellent sample quality that nearly match the gold-standard ensembles generated with expensive metadynamic simulations.

## 2    BACKGROUND AND RELATED WORK

Our work builds on small-molecule conformer generation and protein structure modeling to create a framework for macrocycle conformers. Below, we briefly summarize related work.

**Physics and Heuristic-based Conformer Generation for Macrocycles**    Physics-based and heuristic-based algorithms remain the state of the art for macrocycles and have required special considerations compared to drug-like small molecules due to ring-closing constraints. The open-source cheminformatics library RDKit leverages distance geometry algorithms for small-molecule conformer generation (ETKDG) (Riniker & Landrum, 2015), with improved heuristic bounds for macrocycles (ETKDGv3) (Wang et al., 2020; 2022). Similarly, commercial software such as OpenEye OMEGA (Hawkins et al., 2010; Hawkins & Nicholls, 2012) in `macrocycle` mode uses a distance geometry algorithm based on 4D coordinate initialization to provide diverse conformers (Spellmeyer et al., 1997), as their torsion-driving approach is incompatible with ring closure.

Similarly, low-mode (Kolossváry & Guida, 1996; 1999) or Monte Carlo (Chang et al., 1989) search methods combined with molecular dynamics have been found to be effective at sampling macrocycle conformations, particularly when combined with force field optimizations as demonstrated in Schrödinger's MacroModel (Watts et al., 2014) and Prime MCS (Sindhikara et al., 2017). These

approaches have been tuned with expert knowledge and torsional libraries to maximize agreement with observed experimental structures. The open-source CREST package (Pracht et al., 2020) leverages iterative metadynamics with a genetic structure-crossing algorithm (iMTD-GC) to explore new geometries, and can be considered a gold-standard for generating diverse ensembles of drug-like molecules. In this work, we use the recently-published CREMP (Grambow et al., 2023) dataset, containing high-quality, CREST-generated ensembles, representing over 31 million macrocycle geometries (see Section 4.1 and Appendix B for more details).

One key limitation of these approaches is high computational cost and difficulty in scaling; in general, conformer generation is $10^3 - 10^5 \times$ more computationally expensive compared to a drug-like small molecule due to the increased number of rotatable bonds and their ring-closing constraints (e.g., generating a conformational ensemble of a macrocyclic hexapeptide with CREST requires an average of 14 hours (Grambow et al., 2023)). These approaches become increasingly challenging when kinetic or molecular dynamics approaches are used with explicit solvation (Damjanovic et al., 2021; Linker et al., 2023).

**Generative Approaches for Small Molecule Conformer Ensembles**  Recent work with deep generative models has focused on improved sampling of the conformational landscape of small molecules. For example, Mansimov et al. (2019) propose a conditional graph variational autoencoder (CGVAE) approach for molecular geometry generation. Simm & Hernandez-Lobato (2020) report conditional generation of molecular geometries based on distance geometry. Xu et al. (2021) leverage normalizing flows and energy-based modeling to help capture the multimodal nature and complex dependencies of small molecule space. More recently, Xu et al. (2022) report GeoDiff, an equivariant diffusion-based model that operates on Cartesian point clouds. Although GeoDiff provides strong results, sampling is costly and requires 5,000 time steps. Recently, Zhu et al. (2022) report DMCG, which leverages a variational autoencoder to directly predict coordinates while maintaining invariance to rototranslation and permutation of symmetric atoms.

Recent reports have also drawn inspiration from physics-based conformer generation to leverage the rigid-rotor hypothesis, which treats bond distances and angles as fixed, and torsional angles of rotatable bonds are independently sampled, assuming little or no interdependence between torsions (Schärfer et al., 2013). These include GeoMol (Ganea et al., 2021), an SE(3)-invariant machine learning model for small molecule conformer generation that leverages graph neural networks, and EquiBind (Stärk et al., 2022) which performs conditional generation on protein structure. Recently, Jing et al. (2022) report Torsional Diffusion, a diffusion model that operates on the torsional space via an extrinsic-to-intrinsic score model to provide strong benchmarks on the GEOM dataset (Axelrod & Gómez-Bombarelli, 2022). Importantly, these methods do not address the challenge of highly-coupled torsions within cyclic systems and either propose complex ring-averaging processes (Ganea et al., 2021) or disregard sampling of cyclic structures all together (Jing et al., 2022).

**Protein Structure Prediction and Diffusion**  Significant progress has been made recently in protein structure prediction with the advent of methods such as AlphaFold2 (Jumper et al., 2021) and RoseTTAFold (Baek et al., 2021). However, structure prediction methods have predominantly focused on deterministic maps to static output structures rather than on sampling diverse structure ensembles. Recently, several papers have developed diffusion-based approaches for protein generation based on Euclidean diffusion over Cartesian coordinates (Anand & Achim, 2022; Yim et al., 2023) or backbones as in FoldingDiff (Wu et al., 2022), with an emphasis on structural design. FoldingDiff parameterizes structures over internal backbone angles and torsions and relies on the natural extension reference frame (NeRF) (Parsons et al., 2005) to perform linear reconstructions. However, as we demonstrate below, naive linear transformations fail to address the ring constraints for macrocycles. Moreover, FoldingDiff focuses on *unconditional* generation of protein backbones and is hence not suitable for all-atom conformer ensemble generation. Below, we focus on the challenging problem of conditional generation of constrained geometries.

**Machine Learning Approaches for Macrocycle Conformer Ensemble Generation**  Despite the many approaches focused on small molecules and protein structure generation, there are few efforts in macrocycle structure prediction. Most notably, Miao et al. (2021) recently disclosed StrEAMM for learning on molecular dynamics of cyclic peptides using explicit solvation. StrEAMM is a linear model that predicts local backbone geometries and their respective 1,2- and 1,3-residue interactions to provide excellent ensemble estimates of homodetic hexapeptides, but does not generate explicit all-atom conformers. Additionally, the model is not naturally inductive and is not natively extensible

to other macrocycle ring sizes and residues. Fishman et al. (2023) recently developed a more general framework for diffusion models on manifolds defined via a set of inequality constraints. However, they only investigate the conformational ensemble of a single cyclic peptide as a proof-of-concept using a reduced $\alpha$-carbon representation.

## 3 RINGER: PROBLEM STATEMENT AND METHODS

### 3.1 PROBLEM DEFINITION: CONDITIONAL MACROCYCLE CONFORMER GENERATION

The core objective of our work is to model the distribution of conformers for a macrocyclic peptide with a given amino acid sequence. Given a peptide macrocycle graph $\mathcal{G} = (\mathcal{V}, \mathcal{E})$, where $\mathcal{V}$ is the set of nodes (atoms) and $\mathcal{E}$ is the set of edges (bonds), and $n = |\mathcal{V}|$ our goal is to learn a distribution over the possible conformers. Let $\mathcal{C} = \{c_1, c_2, \ldots, c_K\}$ be the set of conformers, where each conformer $c_k \in \mathcal{C}$ represents a unique spatial arrangement of the atoms $\mathcal{V}$. Our task is to learn the distribution $p(\mathcal{C} \mid \mathcal{G})$, which represents the probability over the conformer ensemble $\mathcal{C}$ given a molecular graph $\mathcal{G}$. Learning and sampling from this complex distribution is inherently challenging for most molecules, and is further complicated in macrocycles due to the highly-coupled nature of ring atoms. A perturbation to one part of the ring generally perturbs the others. Consequently, any model must account for the interdependence between atoms due to the cyclic constraints.

Given this problem, a good generative model ideally satisfies a few key properties: 1) Naturally encodes the physical and structural aspects of macrocyclic peptides. For example, cyclic peptides with only standard peptide bonds (i.e., homodetic peptides) do not have a natural starting residue and hence exhibit cyclic shift invariance, e.g., cyclo-(R.I.N.G.E.R) is identical to cyclo-(I.N.G.E.R.R), where each amino acid is denoted by its one-letter code with "cyclo" indicating cyclization of the sequence. 2) Captures multimodal distributions and complex, higher-order interactions such as the strong coupling between atomic positions in the ring. 3) Samples high-quality and diverse conformations from $p(\mathcal{C} \mid \mathcal{G})$ that faithfully capture realistic geometries while respecting the underlying conformer distribution.

### 3.2 REPRESENTING MACROCYCLE GEOMETRY: REDUNDANT INTERNAL COORDINATES

Conformer geometries are defined by their set of Cartesian coordinates for each atomic position and can hence be modeled using SE(3)-equivariant models to learn complex distributions. However, Euclidean diffusion requires modeling the many degrees of freedom; and, in practice, can require many time steps to generate accurate geometries (Xu et al., 2022). Moreover, realistic conformations are highly sensitive to the precise interatomic distances, angles, and torsions—although this information is implicit in the Cartesian positions, explicitly integrating these quantities into a model can provide a strong inductive bias and accelerate learning (Gasteiger et al., 2020).

Borrowing from molecular geometry optimization (Peng et al., 1996), protein representation (Ramachandran & Sasisekharan, 1968; Dunbrack & Karplus, 1994; Parsons et al., 2005), and inverse kinematics (Han & Rudolph, 2006), we adopt redundant internal coordinates that represent conformer geometries through a set of bond distances, angles, and torsions (dihedral angles), i.e., $\mathcal{C} \equiv \{\mathcal{D}, \Theta, \mathcal{T}\}$. In particular, this simplifies the learning task, as bond distances can be approximated as fixed distances with little loss in accuracy (Hawkins et al., 2010; Wu et al., 2022; Jing et al., 2022), and internal angles typically fit a narrow distribution. Importantly, these coordinates define an internal reference frame that readily encodes complex geometries including chirality. Moreover, this approach obviates the need for complex equivariant networks (Xu et al., 2022; Jing et al., 2022). Hence, our generative process can be reformulated as learning the distribution $p(\{\Theta, \mathcal{T}\} \mid \mathcal{G}; \mathcal{D})$ using fixed bond distances for reconstruction back to Cartesians (Figure 1).

### 3.3 DEEP PROBABILISTIC DIFFUSION MODELS FOR SAMPLING INTERNAL COORDINATES

**Denoising Probabilistic Models** Recent works on deep denoising probabilistic models have demonstrated excellent generative performance for complex multimodal data (Sohl-Dickstein et al., 2015; Ho et al., 2020; Song et al., 2021), and have been successfully applied to both small molecules and proteins (Xu et al., 2022; Wu et al., 2022). We adapt a discrete-time diffusion process (Wu et al., 2022) that formulates the forward transition probability using a wrapped normal distribution, $q(\mathbf{x}_t \mid \mathbf{x}_{t-1}) = \mathcal{N}_{\text{wrapped}}(\mathbf{x}_t; \sqrt{1 - \beta_t}\mathbf{x}_{t-1}, \beta_t \mathbf{I})$, instead of a standard normal distribution (Jing et al., 2022), where $\mathbf{x}_t$ represents the noised internal coordinates (bond angles and torsions) at time step $t$. We train a diffusion model, $p_\Xi(\mathbf{x}_{t-1} \mid \mathbf{x}_t)$, by training a neural architecture, described below,

to predict the noise present at a given time step (for full details, see Appendix C). During inference, we sample $\mathbf{x}_T$ from a wrapped normal distribution and iteratively generate $\mathbf{x}_0$ using $p_\Xi(\mathbf{x}_{t-1} \mid \mathbf{x}_t)$. The sampling process is further detailed in Appendix D.

**Encoder Architecture** Macrocycles exhibit extensive coupling of their residues due to torsional strain and intramolecular interactions such as hydrogen bonds. Here, we use self-attention (Vaswani et al., 2017; Devlin et al., 2019) to learn the complex interactions between atoms. Unlike standard sequence models for linear data, macrocycles exhibit cyclic symmetry with no canonical start position. Thus, we design a modified bidirectional, relative positional encoding (Shaw et al., 2018) $\mathbf{p}_{ij}^K$ to reflect this physical invariance (see Appendix A for notation and Appendix N.4 for ablation study):

$$\mathbf{z}_i = \sum_{j=1}^{n} \alpha_{ij} \left( \mathbf{v}_j \mathbf{W}^V \right), \quad \text{where} \quad \alpha_{ij} = \frac{\exp e_{ij}}{\sum_{k=1}^{n} \exp e_{ik}} \tag{1}$$

$$e_{ij} = \frac{\mathbf{v}_i \mathbf{W}^Q \left( \mathbf{v}_j \mathbf{W}^K + \mathbf{p}_{ij}^K \right)^T}{\sqrt{d_z}} \quad \text{with} \quad \mathbf{p}_{ij}^K = \underbrace{\mathbf{W}_{(i-j) \bmod n}^D}_{\text{forward}} + \underbrace{\mathbf{W}_{(i-j) \bmod (-n)}^D}_{\text{backward}} \tag{2}$$

These cyclic relative position representations specify forward (N-to-C) and reverse (C-to-N) relationships between each atom in the macrocycle, and effectively encode the molecular graph in a self-attention module similar to Ying et al. (2021). The relative position of any neighboring atom is uniquely defined by its forward and reverse graph distances in the embedding lookup $\mathbf{W}^D$. For conditional generation, we perform a linear projection of the atom features $\mathbf{a}_i$, corresponding to *each macrocycle backbone atom and its side chain*, and a separate linear projection of the angles and torsions $\mathbf{x}_i = \boldsymbol{\theta}_i \oplus \boldsymbol{\tau}_i$ and concatenate them as a single input to the transformer, $\mathbf{v}_i = \mathbf{a}_i' \oplus \mathbf{x}_i'$. Notably, our diffusion model only adds noise to the angular component, $\mathbf{x}_i$, corresponding to backbone and side-chain angles and torsions that define a complete atomic configuration. For unconditional backbone generation, atoms are only labeled with their backbone identity (nitrogen, $\alpha$-carbon, carbonyl-carbon) using an embedding that is added to the input, and side chains are not modeled. Model details are shown in Appendix E.

**Ring Closing: Back Conversion to Cartesian Ring Coordinates** Macrocycles with fixed bond distances contain three redundant torsional angles and two redundant bond angles. Whereas linear peptides and proteins can be readily converted into an arbitrary Cartesian reference frame through methods such as NeRF (Parsons et al., 2005), these redundancies prevent direct transformation to unique Cartesians for cyclic structures. Adopting a sequential reconstruction method such as NeRF accumulates small errors that result in inadequate ring closure for macrocycles.[1] Other studies have developed complex heuristics with coordinate averaging for ring smoothing (Ganea et al., 2021), yet these approaches can distort the predicted geometries. In practice, we demonstrate that an efficient post-processing step works well with minimal distortion: we treat this as a constrained optimization problem using the Sequential Least Squares Quadratic Programming (SLSQP) algorithm (Kraft, 1988) to ensure valid Cartesian coordinates while satisfying distance constraints:

$$\hat{\boldsymbol{\xi}} = \arg\min_{\boldsymbol{\xi}} \|\boldsymbol{\theta}(\boldsymbol{\xi}) - \hat{\boldsymbol{\theta}}\|^2 + \|w\left(\boldsymbol{\tau}(\boldsymbol{\xi}) - \hat{\boldsymbol{\tau}}\right)\|^2 \quad \text{subject to:} \quad \mathbf{d}(\boldsymbol{\xi}) = \mathbf{d}_{\text{true}} \tag{3}$$

Here, we find the set of *ring* Cartesian coordinates, $\hat{\boldsymbol{\xi}}$, that minimize the squared error against the ring internal coordinates $\hat{\boldsymbol{\theta}}$ and $\hat{\boldsymbol{\tau}}$ sampled by the diffusion process while satisfying bond distance equality constraints using known ring bond distances, $\mathbf{d}_{\text{true}}$, from the training data. The torsion error, $\boldsymbol{\tau}(\boldsymbol{\xi}) - \hat{\boldsymbol{\tau}}$, is wrapped by $w(\cdot)$ so that it remains in the $[-\pi, \pi)$ range. Empirically, we demonstrate that this scheme recovers realistic macrocycles with high fidelity by evenly distributing the error across the entire macrocycle backbone (see Appendix F for additional details). Additionally, we reject a sample from RINGER if the ring torsion fingerprint deviation (Wang et al., 2020) before versus after optimization with equation 3 is larger than 0.01, which further improves the quality of generated ensembles.

---

[1]Although direct equality and inequality constraints over the diffusion process is a promising direction that could address this problem, we leave this for future work.

**Overall Generation Procedure** Our model represents macrocycles as cyclic sequences of backbone atoms with fixed bond lengths, where each atom is featurized with two ring internal coordinates and several side chain internal coordinates. We train a discrete-time diffusion model to learn a denoising process over the internal coordinates using a transformer architecture with an invariant cyclic positional encoding. Chemical features of the backbone atoms and molecular fingerprints of the attached side-chain atoms provide 2D macrocycle structure information for the sequence-conditioned models. At inference time, we sample from a wrapped Gaussian distribution to produce a set of angles and torsions. In the final post-processing step, macrocycle geometries with Cartesian coordinates are reconstructed through our constrained optimization using equation 3 for ring atoms and using NeRF (Parsons et al., 2005) for rotatable side-chain atoms. Hydrogens and non-rotatable side-chain groups like phenyl rings are not modeled directly and are generated using RDKit. A detailed flowchart illustrating the main steps of our method is shown in Appendix G.

## 4 EXPERIMENTS AND RESULTS

### 4.1 EXPERIMENTAL SETUP

**Dataset** We train and evaluate our approach on the recently published CREMP dataset (Grambow et al., 2023) that contains 36k homodetic macrocyclic peptides across varying ring sizes (4-mers, 5-mers, and 6-mers corresponding to 12-, 15-, and 18-membered backbone rings), side chains, amino-acid stereochemistry, and $N$-methylation. Each macrocycle in CREMP contains a conformational ensemble sampled with CREST (Pracht et al., 2020), a metadynamics algorithm with genetic crossing built on the semi-empirical tight-binding method GFN2-xTB (Bannwarth et al., 2019). We perform stratified random splitting on the data, with a training and validation set of 35,198 molecules (948,158 conformers using a maximum of 30 conformers per molecule), which we split into 90% training and 10% validation, and a final test set of 1,000 molecules corresponding to 877,898 distinct conformers (using *all* conformers per molecule within the $6\,\mathrm{kcal/mol}$ energy threshold defined by CREST). Additional dataset statistics are shown in Appendix B.

**Training & Sampling** All training is performed on the set of 35k peptides described above, using the 30 lowest-energy conformers per peptide. We train each model on a single NVIDIA A100 GPU for up to 1000 epochs until convergence using the Adam optimizer with 10 warmup epochs. Following work in small-molecule conformer generation (Ganea et al., 2021; Xu et al., 2022; Jing et al., 2022), we sample $2K$ conformers for a macrocycle ensemble of $K$ ground-truth conformers (median $K = 656$) and assess them based on the evaluation criteria below. For full training and sampling details see Appendices C and D.

**Evaluation** For unconditional generation, we use Kullback-Leibler divergence to measure the difference in sample quality. For conditional generation, we evaluate the quality of our generated macrocycles using root-mean-squared-deviation (RMSD) between heavy-atom coordinates, similar to previous work on small-molecule conformer generation. We use several metrics including **Mat**ching and **Cov**erage (Xu et al., 2021; Ganea et al., 2021; Jing et al., 2022), and for each we report recall, precision, and F1-score. We also report rRMSD, which is only evaluated on ring atoms. We note that although RMSD is widely used to assess conformer quality, its utility for comparing backbones is more limited, as sampled backbones with highly unrealistic or energetically unfavorable torsions can exhibit low rRMSD values. Therefore, we additionally report the ring torsion fingerprint deviation (rTFD) (Schulz-Gasch et al., 2012; Wang et al., 2020) to evaluate the quality of the torsional profiles. RMSD and rRMSD provide a measure of distance between two conformers based on a least-squares alignment of their respective atomic positions, while rTFD gives a normalized measure of matched torsion angles between backbone geometries. Appendix J defines the evaluation metrics in detail.

**Baselines** We provide benchmarks of our method against open-source and commercial toolkits RDKit ETKDGv3 (for macrocycles) (Wang et al., 2020), OMEGA Macrocycle Mode (OpenEye, 2022), GeoDiff (Xu et al., 2022), DMCG (Zhu et al., 2022), and Torsional Diffusion (Jing et al., 2022). For GeoDiff, DMCG, and Torsional Diffusion we report results trained on CREMP (GeoDiff-Macro, DMCG-Macro, and TorDiff-Macro), as the base models trained on small molecules provided poor performance. To better understand ring diversity, we additionally included a nearest neighbor model (1-NN) based on maximum graph similarity (2D) for ring-evaluation metrics only.

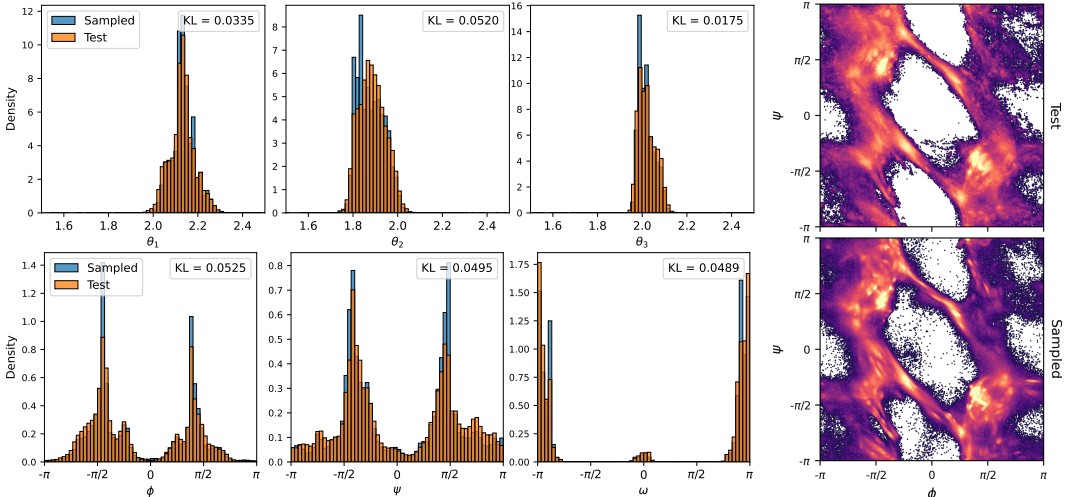

Figure 2: Unconditional conformer generation for a **backbone-only model**. Comparison of the bond angle and dihedral distributions from the held-out test set (orange) and in the *unconditionally* generated redundant internal coordinate samples (blue). The three top left plots correspond to the three bond angle types in each amino acid residue, the three bottom left plots show the three dihedral angles for each residue, and the right shows Ramachandran plots (colored logarithmically by density with high density regions shown in lighter colors). KL divergence is calculated as $D_{KL}(\text{test} \parallel \text{sampled})$.

## 4.2 UNCONDITIONAL GENERATION OF MACROCYCLE BACKBONES

To understand whether this approach can learn the highly-coupled and underlying distribution of macrocycle conformations, we first trained RINGER on macrocycle backbones in the absence of any residue or side-chain features. From a design perspective, diverse backbone sampling alone can help drive inverse peptide design, where specific backbone geometries suggest important sequences. Figure 2 (and per-residue distributions in Appendix L) demonstrates how RINGER accurately replicates both angles and dihedrals with tight fidelity across all residue atoms. Furthermore, we generated Ramachandran plots (Ramachandran & Sasisekharan, 1968) alongside our withheld test set to visualize the conditional dependencies between residue torsions. Notably, RINGER recapitulates the critical modes of the distribution.

## 4.3 CONDITIONAL GENERATION OF MACROCYCLE CONFORMATIONAL ENSEMBLES

We subsequently focused on the challenge of conditional generation to understand whether RINGER could effectively capture the complex steric and intramolecular effects that dictate macrocycle conformation. Whereas the unconditional model disregards side chains, we now condition generation on molecular features corresponding to each atom in the ring, including side-chain information, stereochemistry, and *N*-methylation (see Appendix E) to unambiguously predict all atomic internal coordinates. We evaluate RINGER both in the context of all-atom geometries (RMSD), but also evaluate on backbone-only ring geometries (rRMSD, rTFD) as they are critical for macrocycle design (Bhardwaj et al., 2022).

Comparison of RINGER RMSD, rRMSD, and rTFD ensemble metrics against the baselines are shown in Table 1. Here, recall quantifies the proportion of ground truth conformers that are recovered by the model, and precision quantifies the quality of the generated ensemble. We found that RDKit ETKDGv3 and OMEGA Macrocycle mode, both based on distance-geometry approaches, performed similarly across metrics and achieved moderate recall with limited precision. Comparing to other deep learning approaches, we found that both GeoDiff and DMCG offer improved recall over heuristic baselines, with only DMCG generating improved precision and moderate F1-scores. Moreover, the nearest neighbor baseline, which uses the backbone ensemble from the most similar training molecule (as measured by 2D similarity), provides only modest performance. This baseline demonstrates how side chain identity dictates 3D ensemble geometry, and that deep models cannot trivially memorize backbone geometries in this task. In contrast, RINGER achieves excellent performance across all metrics, providing strong recall and excellent precision over all test molecules, with the best F1-score.

Table 1: *Mean* performance metrics for sequence-conditioned generation of macrocycles. Coverage is evaluated at a threshold of 0.75 Å for all-atom RMSD, 0.1 Å for ring-only RMSD (rRMSD), and 0.05 for ring-only TFD (rTFD). *All test data* conformers are used for evaluation.

| Method | RMSD – Recall | | RMSD – Precision | | RMSD - F1 | |
| --- | --- | --- | --- | --- | --- | --- |
| | COV (%) ↑ | MAT ↓ | COV (%) ↑ | MAT ↓ | COV (%) ↑ | MAT ↓ |
| RDKit (Wang et al., 2020) | 41.1 | 0.853 | 7.5 | 1.419 | 12.7 | 1.065 |
| OMEGA (OpenEye, 2022) | 36.2 | 0.900 | 6.4 | 1.433 | 10.9 | 1.106 |
| GeoDiff-Macro (Xu et al., 2022) | 25.6 | 0.989 | 2.7 | 1.753 | 4.9 | 1.265 |
| DMCG-Macro (Zhu et al., 2022) | **78.3** | **0.534** | 41.9 | 0.942 | 54.6 | 0.682 |
| TorDiff-Macro (Jing et al., 2022) | 44.4 | 0.820 | 8.3 | 1.399 | 14.0 | 1.034 |
| **RINGER** | 62.3 | 0.684 | **76.2** | **0.525** | **68.6** | **0.594** |
| | **rRMSD – Recall** | | **rRMSD – Precision** | | **RMSD - F1** | |
| 1-NN (Seq. Sim.) | 43.7 | 0.301 | 40.3 | 0.331 | 41.9 | 0.315 |
| RDKit (Wang et al., 2020) | 35.8 | 0.187 | 5.6 | 0.540 | 9.7 | 0.277 |
| OMEGA (OpenEye, 2022) | 32.2 | 0.186 | 3.7 | 0.557 | 6.6 | 0.279 |
| GeoDiff-Macro (Xu et al., 2022) | 50.8 | 0.151 | 6.4 | 0.592 | 11.4 | 0.240 |
| DMCG-Macro (Zhu et al., 2022) | **77.6** | **0.076** | 36.6 | 0.301 | 49.8 | 0.121 |
| TorDiff-Macro (Jing et al., 2022) | 35.8 | 0.187 | 5.6 | 0.540 | 9.7 | 0.277 |
| **RINGER** | 76.3 | 0.110 | **81.6** | **0.108** | **78.9** | **0.109** |
| | **rTFD – Recall** | | **rTFD – Precision** | | **RMSD - F1** | |
| 1-NN (Seq. Sim.) | 53.1 | 0.111 | 48.6 | 0.122 | 50.7 | 0.116 |
| RDKit (Wang et al., 2020) | 52.9 | 0.059 | 9.4 | 0.215 | 15.9 | 0.093 |
| OMEGA (OpenEye, 2022) | 49.7 | 0.061 | 6.6 | 0.225 | 11.7 | 0.095 |
| GeoDiff-Macro (Xu et al., 2022) | 68.1 | 0.048 | 9.1 | 0.248 | 16.0 | 0.080 |
| DMCG-Macro (Zhu et al., 2022) | **93.0** | **0.021** | 48.5 | 0.110 | 63.8 | **0.036** |
| TorDiff-Macro (Jing et al., 2022) | 52.8 | 0.059 | 9.4 | 0.215 | 15.9 | 0.093 |
| **RINGER** | 83.8 | 0.035 | **85.3** | **0.038** | **84.5** | **0.037** |

As demonstrated through rRMSD and rTFD metrics, RINGER samples both diverse and high-quality backbone conformations.

## 4.4 Evaluating Conformer Quality with Post Hoc Optimization

The CREMP dataset (Grambow et al., 2023) used in this study leverages the semi-empirical extended tight binding method GFN2-xTB (Bannwarth et al., 2019) for geometry optimization. To enable a level comparison of sampling quality only, we reoptimized all generated samples from all methods with GFN2-xTB (Tables 11 and 12, Appendix O). Notably, we observed a dramatic improvement for xTB-optimized RDKit, OMEGA, GeoDiff-Macro, and TorDiff-Macro molecules, but only slight improvement in precision. In contrast, xTB-reoptimized samples from RINGER see only a minimal but consistent boost in performance, yet still outperform all other approaches. Further analysis of these xTB geometries revealed that RINGER-generated samples require smaller conformational adjustments to reach local minima (Figure 13, Appendix O), suggesting that our model can accurately learn the complex distributions directly from the training data. Overall, these results highlight two key strengths of our machine learning approach: 1) careful featurization enables generating high-quality conformations that closely match quantum chemical geometries, and 2) the diffusion scheme provides good sampling diversity.

## 4.5 Structural Analysis of Generated Macrocycles

Although RMSD and TFD evaluate performance quantitatively, analyzing individual ensembles elucidates the qualitative differences in conformer generation processes. Notably, the two distinct macrocycles in Figure 3 possess ensembles characterized by distinctly unique Ramachandran plots. Importantly, most ground truth conformer ensembles exhibit relatively tight distributions characterized by a distinctive set of $\phi, \psi$ angles (Reference, far left). Although all methods can identify relevant

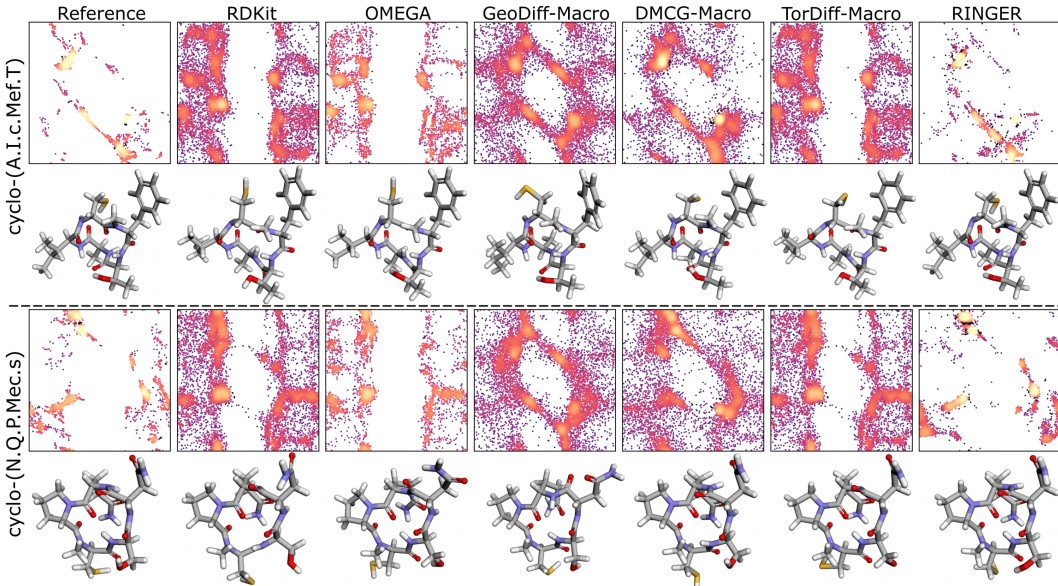

Figure 3: Comparison of macrocycle conformational ensembles. RINGER accurately generates ensembles as illustrated by Ramachandran plots for individual macrocycle ensembles, whereas other methods generate unrealistic backbone geometries. The individual 3D conformers illustrate the lowest-energy reference structure and the closest matching conformer (based on RMSD) from each method.

low-energy conformers, the overall sampling process generates unrealistic distributions for RDKit, OMEGA, and GeoDiff (note that TorDiff relies on RDKit for backbone sampling, and hence produces an identical plot). In contrast, RINGER recapitulates not only the ground state geometry with excellent accuracy, but better captures the entire ensemble distribution. These results demonstrate how RINGER achieves strong performance by providing high sample quality and diversity.

## 5 LIMITATIONS AND FUTURE DIRECTIONS

Our studies demonstrate the potential for diffusion-based models to overcome limitations in constrained macrocycle generation, but they are not without drawbacks. Our current work has focused on the CREMP dataset, which is limited to homodetic, 4-, 5-, and 6-mer macrocycles with canonical side chains. Extension to macrocycles with larger ring sizes, non-canonical side chains, and other complex topologies would improve the generalizability of this work, as well as training on datasets generated with higher levels of theory. Additionally, although we demonstrate the effectiveness of a standard, discrete-time diffusion process, our approach is not physically constrained to satisfy macrocyclic geometries and currently requires a post-optimization step. Developing and applying physics-informed diffusion processes with manifold constraints could improve the efficiency of training and sampling of relevant macrocycle backbones.

## 6 CONCLUSIONS

We present RINGER, a new approach for generating macrocycle conformer ensembles that significantly improves sample quality, diversity, and inference. By leveraging strengths of diffusion-based models, we demonstrate how a transformer-based architecture with a cyclic positional encoding results in significant gains over Cartesian-based equivariant models and widely-used distance geometry-based algorithms for both unconditional and conditional structure generation. The present work paves the way for more efficient and accurate computational exploration of conformational space. We anticipate that this approach will more broadly enable rational macrocycle discovery through further development.

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
