APPENDIX

## A GLOSSARY

Table 2: Glossary of notation and terms used in the methods section.

| Symbol | Description |
|---|---|
| **Molecular Representation and Coordinates** | |
| $\mathcal{G}$ | Macrocycle graph, where $\mathcal{G} = (\mathcal{V}, \mathcal{E})$. |
| $\mathcal{V}$ | Set of atom vertices in a macrocycle graph $\mathcal{G}$. |
| $\mathcal{E}$ | Set of edges (bonds) between the atoms in a macrocycle graph $\mathcal{G}$. |
| $\mathcal{C}$ | Set (ensemble) of conformers for a macrocycle, where $\mathcal{C} = \{c_1, c_2, \ldots, c_K\}$. |
| $\mathcal{D}$ | Bond distances in a conformer ensemble $\mathcal{C}$. |
| $\Theta$ | Bond angles in a conformer ensemble $\mathcal{C}$. |
| $\mathcal{T}$ | Dihedral (torsional) angles in a conformer ensemble $\mathcal{C}$. |
| $\boldsymbol{\xi}$ | Vector of all ring Cartesian coordinates in a conformer. |
| $\mathbf{d}$ | Vector of all ring bond distances in a conformer. |
| $\boldsymbol{\theta}$ | Vector of all ring bond angles in a conformer. |
| $\boldsymbol{\tau}$ | Vector of all ring dihedral (torsional) angles in a conformer. |
| $d_{i,j}$ | Bond distance between atoms $v_i$ and $v_j$. |
| $\theta_{i,j,k}$ | Bond angle between atoms $v_i$, $v_j$, and $v_k$. |
| $\tau_{i,j,k,l}$ | Dihedral (torsional) angle between atoms $v_i$, $v_j$, $v_k$, and $v_l$. |
| $\phi$ | Dihedral angle of bond between nitrogen and $\alpha$-carbon. |
| $\psi$ | Dihedral angle of bond between $\alpha$-carbon and carbonyl-carbon. |
| $\omega$ | Dihedral angle of bond between carbonyl-carbon and nitrogen (peptide bond). |
| **Encoder Model** | |
| $\mathbf{a}_i$ | Atom features for ring vertex $v_i$. |
| $\boldsymbol{\theta}_i$ | Bond angles corresponding to ring vertex $v_i$ (1 ring & 5 side chain). |
| $\boldsymbol{\tau}_i$ | Dihedral (torsional) angles corresponding to ring vertex $v_i$ (1 ring & 5 side chain). |
| $\mathbf{x}_i$ | Internal coordinates corresponding to vertex $v_i$, where $\mathbf{x}_i = \boldsymbol{\theta}_i \oplus \boldsymbol{\tau}_i$. |
| $\mathbf{v}_i$ | Input/hidden representation for ring vertex $v_i$. |
| $\mathbf{z}_i$ | Self-attention output for vertex $v_i$. |
| $\alpha_{ij}$ | Attention probability between vertices $v_i$ and $v_j$. |
| $e_{ij}$ | Unnormalized attention score between vertices $v_i$ and $v_j$. |
| $d_z$ | Attention head dimensionality. |
| $\mathbf{p}_{ij}^K$ | Cyclic relative positional embedding between vertices $v_i$ and $v_j$. |
| $\mathbf{W}^K$ | Key matrix. |
| $\mathbf{W}^Q$ | Query matrix. |
| $\mathbf{W}^V$ | Value matrix. |
| $\mathbf{W}^D$ | Graph-distance embedding matrix. |
| **Diffusion Process** | |
| $\mathbf{x}_t$ | Noised internal coordinates (bond angles and torsions) at time step $t$. |
| $q(\mathbf{x}_t \mid \mathbf{x}_{t-1})$ | Forward transition probability. |
| $p_\Xi(\mathbf{x}_{t-1} \mid \mathbf{x}_t)$ | Diffusion model (reverse transition probability) parameterized by $\Xi$. |
| $\beta_t$ | Variance from cosine variance schedule at time step $t$. |
| $\boldsymbol{\epsilon}_t$ | Noise scale at time step $t$. |
| **Miscellaneous** | |
| $\hat{\cdot}$ | Denotes predicted/generated quantity. |
| $w(\cdot)$ | Function to wrap within $[-\pi, \pi)$ range, $w(\tau) := (\tau + \pi) \mod (2\pi) - \pi$. |
| $\delta$ | Threshold for evaluating Coverage metric. |
| $\oplus$ | Concatenation. |

# B  DATASET DESCRIPTION

We leverage the recently described Conformer-Rotamer Ensembles of Macrocyclic Peptides (CREMP) dataset (Grambow et al., 2023) that contains 36,198 unique macrocyclic peptide sequences and their corresponding ensembles, totaling 31.3 million conformers. All conformers in CREMP were optimized using the GFN2-xTB semi-empirical quantum chemistry method (Bannwarth et al., 2019). GFN2-xTB incorporates physics-based terms for dispersion, electrostatics, hydrogen bonding, and other quantum mechanical effects into a self-consistent tight-binding framework. This provides a balance of reasonable accuracy and computational efficiency, bridging the gap between fast but inaccurate force fields and high-level yet expensive DFT methods.

The conformational sampling was performed using CREST (Pracht et al., 2020), which combines metadynamics enhanced sampling with GFN2-xTB for energies and forces. Metadynamics iteratively adds biasing potentials to guide sampling to unexplored areas of conformational space, enabling more thorough sampling than conventional molecular dynamics. By pairing metadynamics with GFN2-xTB, CREST balances accuracy and computational efficiency when generating macrocycle ensembles. However, metadynamics remains expensive, requiring thousands of CPU hours per molecule. The CREMP dataset hence provides extensive high-quality training data of macrocycle conformers. Our work builds on CREMP, by developing deep generative models to approximate these computationally demanding physics-based methods for sampling.

Table 3: Dataset statistics for CREMP (Grambow et al., 2023).

| Residues | Molecules | Conformers | | | | | |
|---|---|---|---|---|---|---|---|
| | | Count | Mean | Median | Std. Dev. | Min. | Max. |
| 4 | 17,842 | 12,205,128 | 684 | 508 | 677 | 1 | 12,268 |
| 5 | 13,644 | 14,134,609 | 1,036 | 825 | 824 | 6 | 8,486 |
| 6 | 4,712 | 4,921,068 | 1,044 | 879 | 764 | 28 | 5,619 |
| Total | 36,198 | 31,260,805 | 864 | 656 | 768 | 1 | 12,268 |

# C  TRAINING DETAILS

We adapt a discrete-time diffusion scheme that formulates the forward transition probability using a wrapped normal distribution,

$$
q\left(\mathbf{x}_t \mid \mathbf{x}_{t-1}\right) = \mathcal{N}_{\text{wrapped}}\left(\mathbf{x}_t; \sqrt{1-\beta_t}\mathbf{x}_{t-1}, \beta_t\mathbf{I}\right)
$$
$$
= \frac{1}{\beta_t\sqrt{2\pi}} \sum_{\mathbf{k}\in\mathbb{Z}^n} \exp\left(\frac{-\left\|\mathbf{x}_t - \sqrt{1-\beta_t}\mathbf{x}_{t-1} + 2\pi\mathbf{k}\right\|^2}{2\beta_t^2}\right) \tag{4}
$$

instead of a standard normal distribution (Wu et al., 2022; Jing et al., 2022), where $\mathbf{x}_t$ represents the noised internal coordinates (bond angles and torsions) at time step $t$. The diffusion model, $p_\Xi(\mathbf{x}_{t-1} \mid \mathbf{x}_t)$, parameterized by $\Xi$, reverses the process to denoise a wrapped normal distribution toward the data distribution. In the conditional setting, we further guide the diffusion process by learning $p_\Xi(\mathbf{x}_{t-1} \mid \mathbf{x}_t, \mathcal{G})$ in order to draw samples from the ensemble for a specific macrocycle, $\mathcal{G}$. We use the same cosine variance schedule as Wu et al. (2022) and Nichol & Dhariwal (2021) for $\beta_t \in (0,1)_{t=1}^T$, but with significantly fewer time steps ($T = 20$). $p_\Xi(\mathbf{x}_{t-1} \mid \mathbf{x}_t)$ and $p_\Xi(\mathbf{x}_{t-1} \mid \mathbf{x}_t, \mathcal{G})$ are trained using the simplified objective from Ho et al. (2020) to train a neural network, $\boldsymbol{\epsilon}_\Xi(\mathbf{x}_t, t)$, to predict the noise present at a given time step by minimizing a smooth L1 loss (Girshick, 2015) wrapped by $w(\mathbf{x}) = (\mathbf{x} + \pi) \bmod (2\pi) - \pi$:

$$
\mathbf{d}_w = w\left(\boldsymbol{\epsilon} - \boldsymbol{\epsilon}_\Xi\left(w\left(\sqrt{\bar{\alpha}_t}\mathbf{x}_0 + \sqrt{1-\bar{\alpha}_t}\boldsymbol{\epsilon}\right), t\right)\right)
$$
$$
L_w = \frac{1}{N} \sum_{i=1}^N \begin{cases} 0.5\frac{d_{w,i}^2}{\beta_L} & \text{if } |d_{w,i}| < \beta_L \\ |d_{w,i}| - 0.5\beta_L & \text{otherwise} \end{cases} \tag{5}
$$

with $\beta_L = 0.1\pi$ as the transition point between L1 and L2 regimes (Wu et al., 2022), $\alpha_t = 1 - \beta_t$, and $\bar{\alpha}_t = \prod_{s=1}^{t} \alpha_s$. We sample time steps uniformly from $t \sim U(0, T)$ during training and shift the bond angles and dihedrals using the element-wise means from the training data.

# D  SAMPLING DETAILS

During inference, we first sample $\mathbf{x}_T$ from a wrapped normal distribution and iteratively generate $\mathbf{x}_0$ from $t = T$ to $t = 1$ using

$$\mathbf{x}_{t-1} = w \left( \frac{1}{\sqrt{\alpha_t}} \left( \mathbf{x}_t - \frac{1 - \alpha_t}{\sqrt{1 - \bar{\alpha}_t}} \boldsymbol{\epsilon}_\Xi(\mathbf{x}_t, t) \right) + \sigma_t \mathbf{x} \right) \tag{6}$$

where $\sigma_t = \sqrt{\beta_t (1 - \bar{\alpha}_{t-1})/(1 - \bar{\alpha}_t)}$ is the variance of the reverse process and $\mathbf{z} = \mathcal{N}(\mathbf{0}, \mathbf{I})$ if $t > 1$ and $\mathbf{z} = \mathbf{0}$ otherwise (Wu et al., 2022).

# E  MODEL DETAILS AND HYPERPARAMETERS

Our model is a BERT transformer (Devlin et al., 2019) with graph-based, cyclic relative positional encodings described in equation 1 and equation 2. The model input is a sequence of internal coordinates (and atom features for the conditional model). We linearly upscale the model input (bond angles and dihedrals) and separately upscale the atom features. Angles and atom features are then concatenated. The time step is embedded using random Fourier embeddings (Song et al., 2021) and added to the upscaled input. The combined embeddings are passed through the BERT transformer, the output of which is passed through a two-layer feed-forward network with GELU activation and layer normalization. Relevant hyperparameters are shown in Table 4.

Table 4: Hyperparameters.

| Parameter | Value |
|---|---|
| Angle embedding size | 256 |
| Atom feature embedding size | 128 |
| Encoder layer dimensionality (hidden size) | 384 |
| Number of hidden layers | 12 |
| Number of attention heads | 12 |
| Feed-forward layer dimensionality (intermediate size) | 512 |
| Optimizer | AdamW |
| Learning rate | $10^{-3}$ (restarted with $5 \times 10^{-4}$) |
| Maximum number of epochs | 1000 |
| Warmup epochs | 10 |
| Batch size | 8192 |

To condition on the atom sequence, we encode each atom using features of the atom itself and a Morgan fingerprint representation of the side chain attached to the atom (including the atom itself). The atom features include the atomic number, a chiral tag (L, D, or no chirality), aromaticity, hybridization, degree, valence, number of hydrogens, charge, sizes of rings that the atom is in, and the number of rings that the atom is in. The Morgan fingerprint is a count fingerprint with radius 3 and size 32.

# F   OPTIMIZATION FOR BACK CONVERSION TO CARTESIAN RING COORDINATES

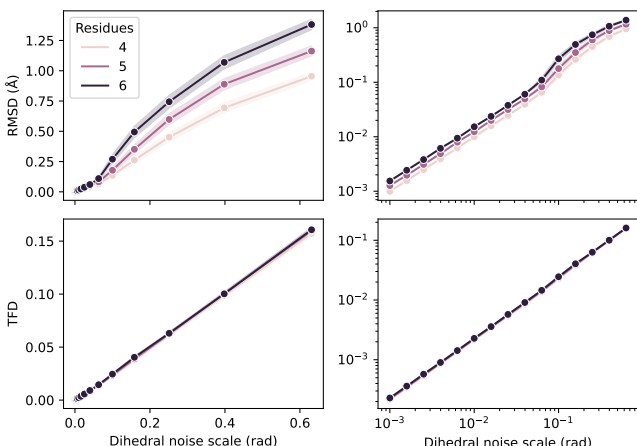

Figure 4: Our constrained optimization procedure is robust to noise as illustrated by a synthetic test of applying noise to the dihedral angles, recovering Cartesian coordinates using equation 3, and comparing to the initial geometry in terms of rRMSD and rTFD.

To convert from the set of redundant internal coordinates predicted by the model back to Cartesian coordinates, we solve the optimization in equation 3 to obtain a set of Cartesian coordinates that exactly satisfies the known bond distances in the ring. To demonstrate that this procedure is robust to noise, we repeatedly embed 4-, 5-, and 6-mer backbones in 3D using RDKit distance geometry, extract their (redundant) internal coordinates, and add noise to the dihedral angles at different noise scales (standard deviation of a normal distribution) while ensuring that angles always remain in the $[-\pi, \pi)$ range. This creates a set of inconsistent, redundant dihedral angles, i.e., there exists no direct correspondence in Cartesian coordinates. We recover a possible Cartesian configuration using equation 3 and compute rRMSD and rTFD for the ring atoms compared to the "true" internal coordinates from the RDKit geometry. Figure 4 shows that even moderate errors ($\sim 0.1\,\mathrm{rad}$) result in very small errors in terms of both rRMSD ($\sim 0.1\,\text{Å}$) and rTFD ($\sim 0.02$).

Notably, the optimization problem in equation 3 is non-convex and requires a suitable initial guess to perform well. We assign this initial guess by obtaining a Cartesian geometry using the approach of sequentially setting atom positions according to the sequence of bond distances, angles, and torsions starting from each of the atoms in the ring. We then average the so-generated $n_{\mathrm{ring}}$ sets of Cartesian coordinates and use the resulting coordinates as the initial guess.

# G  OVERALL METHOD OVERVIEW

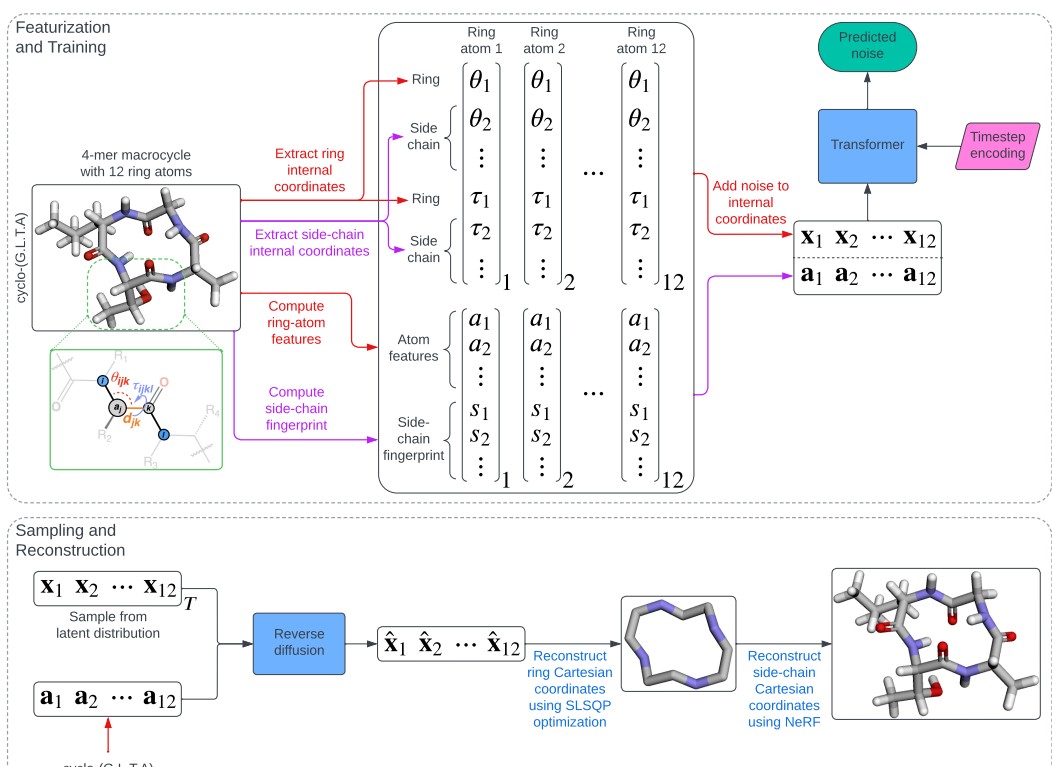

Figure 5: Overall training and sampling procedure.

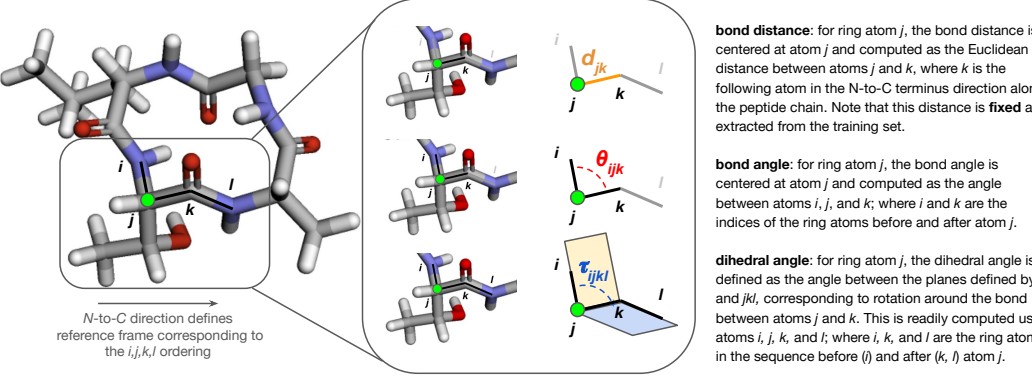

Figure 6: **Schematic representation of how internal coordinates are assigned to each ring atom.** For each atom in the macrocycle, we define a consistent direction based on the N-to-C directionality of the peptide backbone. Neighboring atoms then define the key vectors and planes used to calculate bond distances, angles, and torsions. Specifically, for an atom $j$, the N-to-C direction defines a set of indices $(i, j, k, l)$ where $i = j - 1$, $k = j + 1$, and $l = j + 2$ along the macrocycle backbone. Internal coordinates are readily calculated from their Cartesian coordinates through standard geometry calculations.

## H SOFTWARE

All experiments were performed using Python and standard numerical libraries. For cheminformatics analysis, all molecules were processed using either OpenEye Applications and Toolkits (OpenEye, 2022) or the open-source cheminformatics library RDKit (Landrum, 2006). We implemented all experiments in Python using PyTorch (Paszke et al., 2019) and PyTorch Lightning (Falcon & The PyTorch Lightning team, 2019). Transformers were implemented using BERT models within HuggingFace Transformers (Wolf et al., 2020).

## I HARDWARE

Each model was trained on a single NVIDIA A100 GPU with 80 GB VRAM using 12 CPUs for data loading and 96 GB of memory.

## J EVALUATION

To measure both diversity and quality of the generated ensembles, we follow previous work and leverage four RMSD-based metrics (Xu et al., 2021; Ganea et al., 2021). The *recall*-based **Cov**erage metric measures the percentage of correctly generated conformers at a certain RMSD threshold, $\delta_{\text{RMSD}}$. For a ground-truth ensemble $\mathcal{C}$ and a generated ensemble $\hat{\mathcal{C}}$:

$$\text{RMSD-COV-R}(\hat{\mathcal{C}}, \mathcal{C}) = \frac{1}{|\mathcal{C}|} \left| \left\{ c \in \mathcal{C} : \exists \hat{c} \in \hat{\mathcal{C}}, \text{RMSD}(\hat{c}, c) \leq \delta_{\text{RMSD}} \right\} \right| \tag{7}$$

The *recall*-based **Mat**ching metric measures the average RMSD across the closest-matching (minimum-RMSD) generated conformer for each ground-truth conformer:

$$\text{RMSD-MAT-R}(\hat{\mathcal{C}}, \mathcal{C}) = \frac{1}{|\mathcal{C}|} \sum_{c \in \mathcal{C}} \min_{\hat{c} \in \hat{c}} \text{RMSD}(\hat{c}, c) \tag{8}$$

The other two RMSD-based metrics are *precision* metrics that are defined identically, except that the ground-truth and generated ensembles are switched, and therefore constitute a measure of how many generated conformers are of high quality. Similarly, we compute four RMSD metrics on the ring atoms *only*, indicated using rRMSD.

Analogous to the RMSD-based metrics, we define four metrics based on ring torsion fingerprint deviation (rTFD) (Schulz-Gasch et al., 2012; Wang et al., 2020) to measure diversity and quality in terms of the torsional profiles of the generated rings:

$$\text{rTFD-COV-R}(\hat{\mathcal{C}}, \mathcal{C}) = \frac{1}{|\mathcal{C}|} \left| \left\{ c \in \mathcal{C} : \exists \hat{c} \in \hat{\mathcal{C}}, \text{rTFD}(\hat{c}, c) \leq \delta_{\text{rTFD}} \right\} \right| \tag{9}$$

$$\text{rTFD-MAT-R}(\hat{\mathcal{C}}, \mathcal{C}) = \frac{1}{|\mathcal{C}|} \sum_{c \in \mathcal{C}} \min_{\hat{c} \in \hat{c}} \text{rTFD}(\hat{c}, c) \tag{10}$$

rTFD quantifies how well the macrocycle torsion angles match between two conformers and is given by (Wang et al., 2020):

$$\text{rTFD}(\hat{c}, c) = \frac{1}{n_{\text{ring}}} \sum_{i=1}^{n_{\text{ring}}} \frac{1}{\pi} \left| w\left(\tau_i(\hat{c}) - \tau_i(c)\right) \right| \tag{11}$$

where $\tau_i(c)$ extracts the $i$-th macrocycle torsion angle of conformer $c$ and $w(\cdot)$ ensures that the deviation is wrapped correctly around the $[-\pi, \pi)$ boundary. Each torsion deviation is normalized by the maximum (absolute) deviation, $\pi$, so that rTFD lies in $[0, 1]$.

For both COV and MAT we also compute an F1 score, which is defined as the harmonic mean between precision and recall.

## K Conformer Generation Baselines

**RDKit ETKDGv3**   RDKit baselines used ETKDGv3 (Riniker & Landrum, 2015; Wang et al., 2020) with macrocycle torsion preferences. We first embedded up to $2K$ conformers (where $K$ is the number of true conformers) using `EmbedMultipleConfs` with random coordinate initialization (`useRandomCoords=True`), which has been shown to be beneficial for generating macrocycle geometries (Wang et al., 2020). Conformers were subsequently optimized using MMFF94 (Halgren, 1996) as implemented in RDKit and sorted by energy. Finally, the sorted conformers were filtered based on heavy-atom RMSD with a threshold of 0.1 Å.

**OpenEye OMEGA: Macrocycle Mode**   OMEGA baselines were performed using OpenEye Applications (`2022.1.1`) with OMEGA (`v.4.2.0`) (Hawkins et al., 2010; Hawkins & Nicholls, 2012) in `macrocycle` mode (Spellmeyer et al., 1997). Conformational ensembles were generated with the following `macrocycle` settings: `maxconfs=2K`, `ewindow=20`, `rms=0.1`, `dielectric_constant=5.0`, where $K$ corresponds to the number of ground truth conformers from the original CREST ensemble in the CREMP dataset. The dielectric constant was set to 5.0 (chloroform) to most closely mimic the implicit chloroform solvation used in CREMP.

**GeoDiff-Macro**   We used the original paper implementation and code of GeoDiff from Xu et al. (2022) available at https://github.com/MinkaiXu/GeoDiff, which we retrained to convergence on the CREMP dataset with the same data splits. We used the same experimental details as the GEOM-Drugs model from Xu et al. (2022). As with all the other methods, we evaluated GeoDiff by sampling $2K$ conformers for each molecule. Inference for GeoDiff uses 5,000 time steps, which required more than 24 h for all test set molecules on 20 A100 GPUs.

**DMCG-Macro**   We used the original paper implementation and code of DMCG from Zhu et al. (2022) available at https://github.com/DirectMolecularConfGen/DMCG, which we retrained to convergence on the CREMP dataset with the same data splits. After a basic hyperparameter search, we used the same experimental details as the GEOM-Drugs model from Zhu et al. (2022). Regardless of trainer hyperparameters, strong overfitting occurred after two cycles through the cyclic learning rate scheduler based on the validation loss. Therefore, we selected the best checkpoint by evaluating the first two validation loss minima using the metrics described in Appendix J and selecting the best one. It should be noted that the DMCG model has an *order of magnitude more parameters* than the other methods. As with all the other methods, we evaluated DMCG by sampling $2K$ conformers for each molecule.

**TorDiff-Macro**   We used the original paper implementation and code of TorDiff from Jing et al. (2022) available at https://github.com/gcorso/torsional-diffusion, which we retrained to convergence on the CREMP dataset with the same data splits. We used the same experimental details as the GEOM-Drugs model from Jing et al. (2022) with slight modifications to improve the performance for the macrocycle structure task: Prior to training, a conformer matching procedure is necessary to account for the distributional shift between RDKit local structures and xTB-optimized geometries. In order to account for this, we seeded the training local structures using the ground-truth xTB-optimized conformers with subsequent MMFF94 force-field optimization. At inference time, we seeded the local structures by generating RDKit conformers as described above, which very significantly boosted performance of TorDiff. The original implementation from Jing et al. (2022) only seeds conformers using default RDKit distance geometry embedding parameters, which generates low-quality conformers for macrocycles and leads to very poor performance.

**1-NN (Nearest Neighbor Baseline)**   As an instance-based baseline for evaluating macrocycle backbones only (with rRMSD and rTFD), we use a simple 2D similarity approach to find the nearest sequence neighbor for a test molecule within the training set. Each macrocycle is featurized using a residue-wise RDKit Morgan fingerprint, and we calculate the maximum similarity by exhaustively comparing all possible sequence alignments across each training set macrocycle. The backbone is

then extracted from the training set molecules and its conformers are used for ring evaluation as above.

## L  UNCONDITIONAL BACKBONE GENERATION: ADDITIONAL RESULTS AND PLOTS

As a proof-of-concept, we trained a backbone-only unconditional model to understand whether our approach could accurately model the complex distribution of coupled coordinates. We provide additional plots below, split by macrocycle size, to demonstrate that our model is expressive enough to capture the critical modes of these distributions with granularity.

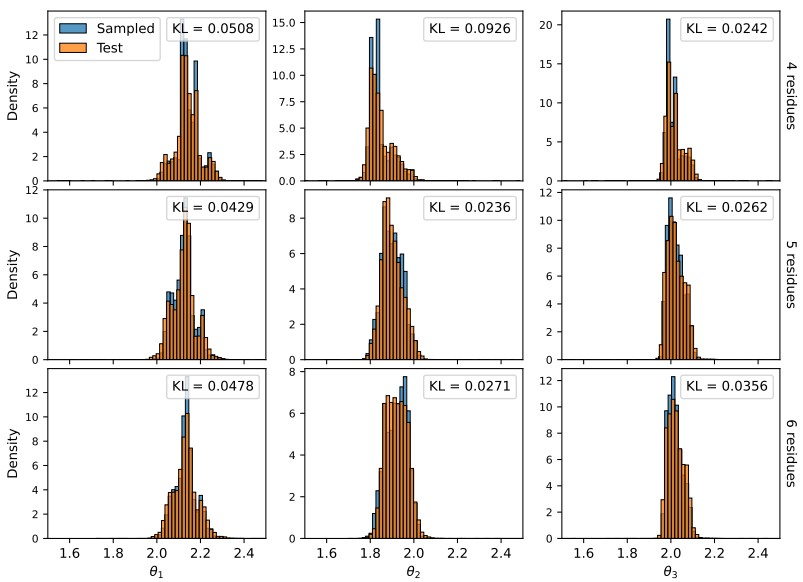

Figure 7: Bond angle distributions split by number of residues for the backbone-only unconditional model.

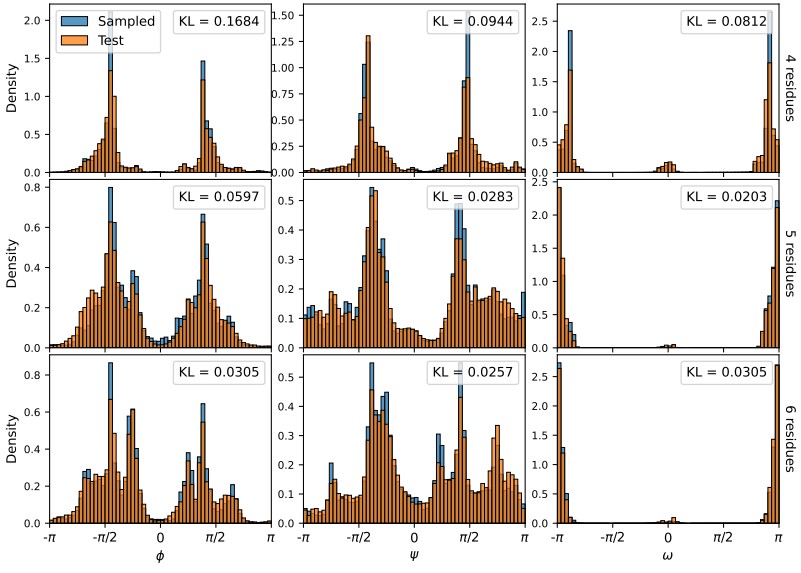

Figure 8: Dihedral angle distributions split by number of residues for the backbone-only unconditional model.

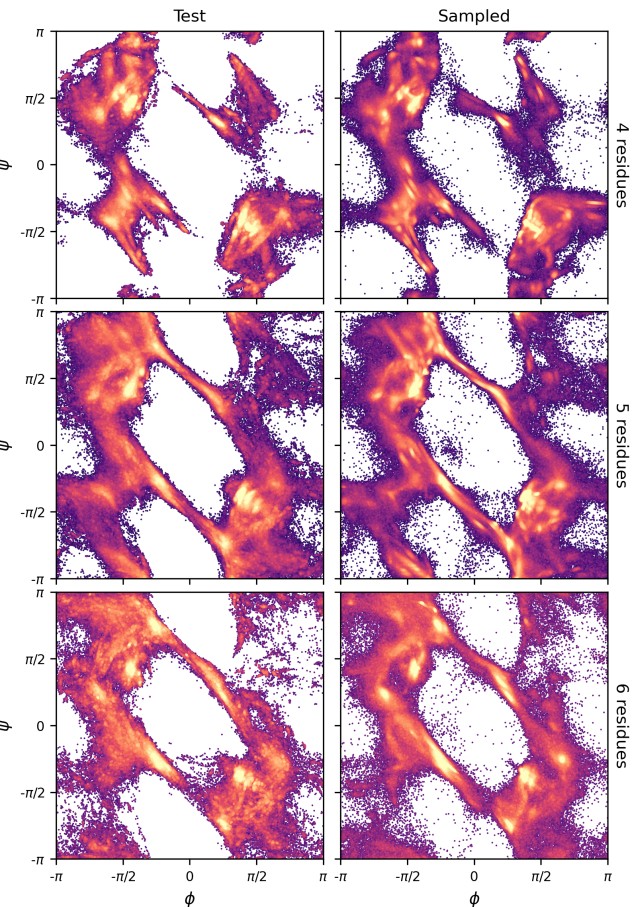

Figure 9: Ramachandran distributions split by number of residues for the backbone-only unconditional model.

## M  SEQUENCE-CONDITIONED GENERATION: BACKBONE-ONLY MODEL RESULTS AND PLOTS

We initially investigated sequence-conditioned generation for a backbone-only model to better understand the effect of key hyperparameter choices (e.g. timesteps, number of training set conformers, etc). For completeness, we include the results of these preliminary studies below.

Table 5: Performance metrics for sequence-conditioned generation of macrocycles evaluated on ring atoms. Coverage is evaluated at a threshold of $0.1$ Å for rRMSD and $0.05$ for rTFD. $k$ is the maximum number of lowest-energy conformers used per molecule in the *training* data. *All test data* conformers are used for evaluation. "opt" refers to the use of equation 3 to reconstruct Cartesian coordinates.

| | | rRMSD – Recall | | | | rRMSD – Precision | | | |
| | | Coverage (%) ↑ | | MAT (Å) ↓ | | Coverage (%) ↑ | | MAT (Å) ↓ | |
| Method | $k$ | Mean | Med. | Mean | Med. | Mean | Med. | Mean | Med. |
|---|---|---|---|---|---|---|---|---|---|
| RDKit (Wang et al., 2020) | – | 35.8 | 8.9 | 0.187 | 0.160 | 5.6 | 0.9 | 0.540 | 0.504 |
| OMEGA (OpenEye, 2022) | – | 32.3 | 7.1 | 0.186 | 0.163 | 3.7 | 1.3 | 0.557 | 0.525 |
| GeoDiff-Macro (Xu et al., 2022) | 30 | 50.8 | 54.2 | 0.151 | 0.120 | 6.4 | 3.0 | 0.592 | 0.559 |
| **RINGER** | 30 | 77.0 | 84.5 | 0.091 | 0.072 | **61.3** | **69.1** | **0.185** | **0.120** |
| **RINGER** (opt) | 1 | 63.8 | 66.9 | 0.139 | 0.112 | 58.1 | 65.1 | 0.430 | 0.327 |
| **RINGER** (opt) | 30 | 79.7 | 86.3 | 0.084 | 0.065 | 56.4 | 62.7 | 0.441 | 0.356 |
| **RINGER** (opt) | 100 | **85.6** | **92.2** | **0.065** | **0.049** | 56.9 | 62.4 | 0.454 | 0.385 |

| | | rTFD – Recall | | | | rTFD – Precision | | | |
| | | Coverage (%) ↑ | | MAT ↓ | | Coverage (%) ↑ | | MAT ↓ | |
| Method | $k$ | Mean | Med. | Mean | Med. | Mean | Med. | Mean | Med. |
|---|---|---|---|---|---|---|---|---|---|
| RDKit (Wang et al., 2020) | – | 52.9 | 55.3 | 0.059 | 0.051 | 9.4 | 4.4 | 0.215 | 0.206 |
| OMEGA (OpenEye, 2022) | – | 49.7 | 47.6 | 0.061 | 0.055 | 6.6 | 4.2 | 0.225 | 0.219 |
| GeoDiff-Macro (Xu et al., 2022) | 30 | 68.1 | 83.0 | 0.048 | 0.037 | 9.1 | 6.1 | 0.248 | 0.241 |
| **RINGER** | 30 | **90.1** | **95.0** | **0.024** | **0.019** | **74.7** | **86.2** | **0.059** | **0.033** |
| **RINGER** (opt) | 30 | 89.2 | 94.3 | **0.024** | **0.019** | 61.8 | 68.9 | 0.068 | 0.044 |

Table 6: Evaluating RINGER (opt) trained and sampled with different numbers of timesteps.

| | rRMSD – Recall | | | | rRMSD – Precision | | | |
| | Coverage (%) ↑ | | MAT (Å) ↓ | | Coverage (%) ↑ | | MAT (Å) ↓ | |
| Timesteps | Mean | Med. | Mean | Med. | Mean | Med. | Mean | Med. |
|---|---|---|---|---|---|---|---|---|
| 20 | 79.7 | 86.3 | 0.084 | 0.065 | 56.4 | 62.7 | 0.441 | 0.356 |
| 50 | 80.8 | 88.0 | 0.082 | 0.061 | **60.5** | **68.9** | **0.431** | **0.335** |
| 100 | **81.5** | **88.9** | **0.080** | **0.060** | 58.0 | 65.2 | 0.443 | 0.365 |

Table 7 shows that bond angles are required in addition to dihedral angles in order for the model to perform well. To reconstruct Cartesian geometries using the dihedral-only model, we modified equation 3 to include inequality constraints for the bond angles where the upper and lower limit are determined by the standard deviations of bond angles from the training data.

Table 7: Evaluating RINGER (opt) trained only with dihedral angles.

| | rRMSD – Recall | | | | rRMSD – Precision | | | |
| --- | --- | --- | --- | --- | --- | --- | --- | --- |
| | Coverage (%) ↑ | | MAT (Å) ↓ | | Coverage (%) ↑ | | MAT (Å) ↓ | |
| | Mean | Med. | Mean | Med. | Mean | Med. | Mean | Med. |
| $\mathbf{x}_i = [\theta_i, \tau_i]$ | **79.7** | **86.3** | **0.084** | **0.065** | **56.4** | **62.7** | **0.441** | **0.356** |
| $\mathbf{x}_i = [\tau_i]$ | 66.8 | 73.5 | 0.130 | 0.101 | 41.1 | 37.6 | 0.469 | 0.417 |
| | rTFD – Recall | | | | rTFD – Precision | | | |
| | Coverage (%) ↑ | | MAT ↓ | | Coverage (%) ↑ | | MAT ↓ | |
| | Mean | Med. | Mean | Med. | Mean | Med. | Mean | Med. |
| $\mathbf{x}_i = [\theta_i, \tau_i]$ | **89.2** | **94.3** | **0.024** | **0.019** | **61.8** | **68.9** | **0.068** | **0.044** |
| $\mathbf{x}_i = [\tau_i]$ | 83.2 | 91.0 | 0.035 | 0.024 | 49.2 | 49.3 | 0.144 | 0.114 |

# N    SEQUENCE-CONDITIONED GENERATION: ADDITIONAL RESULTS AND PLOTS

## N.1    DISTRIBUTIONS

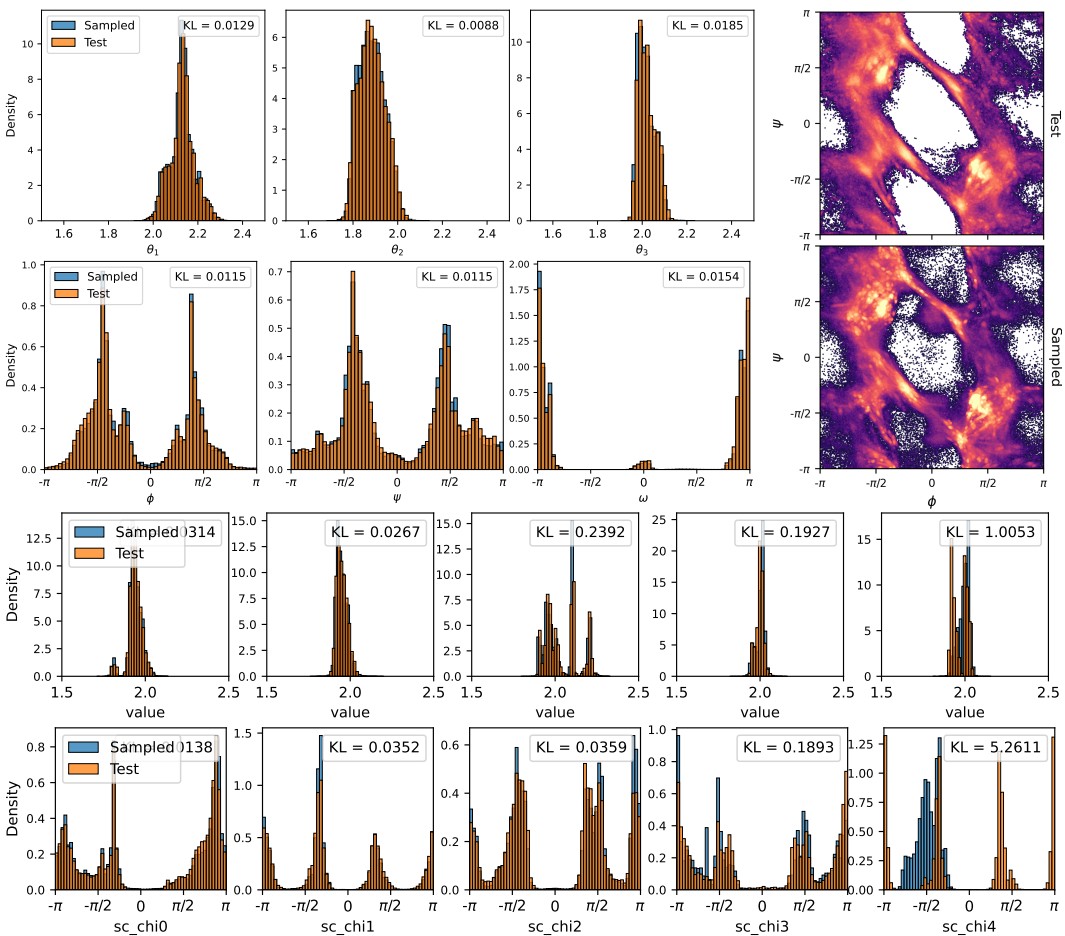

Figure 10: Comparison of the bond angle and dihedral distributions from the held-out test set and in the *conditionally* generated samples (prior to reconstruction to Cartesian coordinates). The top two rows show the distributions of internal coordinates in the ring and the bottom two rows show the side-chain distributions.

Figure 11 shows the Ramachandran plots split by number of residues for the conditional model and illustrates the effect of equation 3 to reconstruct realizable Cartesian geometries from the set of redundant internal coordinates predicted by the model. Notably, while the reconstructed geometries still reproduce the joint distribution over dihedral angles well, several artifacts are introduced as a result of the optimization, which motivates the further development of generative methods that directly incorporate the cyclic constraints into the diffusion process itself.

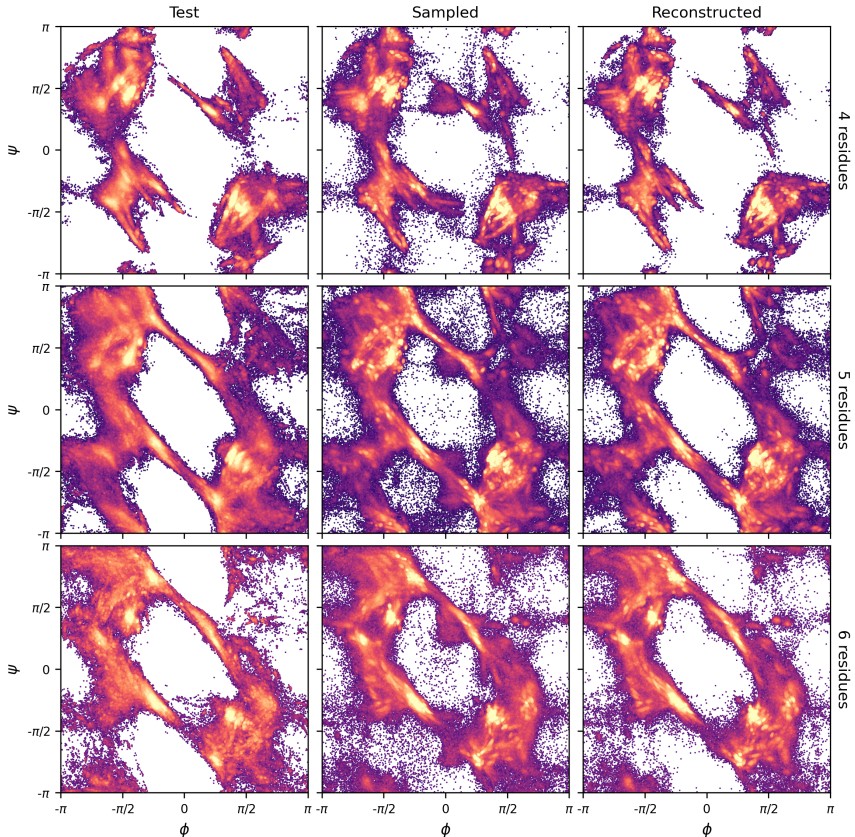

Figure 11: Ramachandran distributions for *conditionally* generated samples split by number of residues. The "Reconstructed" column shows the distributions after converting to Cartesian coordinates using the SLSQP optimization in equation 3 followed by rejection of inadequate samples.

## N.2 COVERAGE

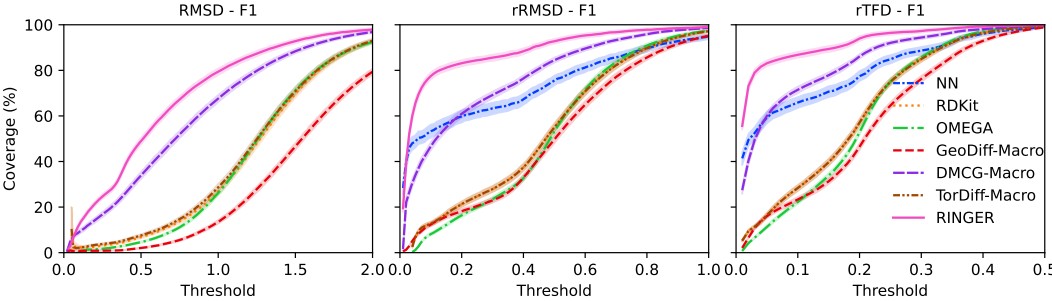

Figure 12: Comparison of mean coverage when varying the threshold. Translucent error bands correspond to 95% confidence intervals.

## N.3   MEDIAN PERFORMANCE METRICS

Table 8: *Median* performance metrics for sequence-conditioned generation of macrocycles. Coverage is evaluated at a threshold of 0.75 Å for all-atom RMSD, 0.1 Å for ring-only RMSD (rRMSD), and 0.05 for ring-only TFD (rTFD). *All test data* conformers are used for evaluation.

| Method | RMSD – Recall | | RMSD – Precision | | RMSD - F1 | |
|---|---|---|---|---|---|---|
| | COV (%) ↑ | MAT ↓ | COV (%) ↑ | MAT ↓ | COV (%) ↑ | MAT ↓ |
| RDKit (Wang et al., 2020) | 33.0 | 0.830 | 3.7 | 1.357 | 6.7 | 1.030 |
| OMEGA (OpenEye, 2022) | 31.3 | 0.852 | 4.9 | 1.360 | 8.5 | 1.047 |
| GeoDiff-Macro (Xu et al., 2022) | 14.3 | 0.949 | 1.1 | 1.701 | 2.0 | 1.218 |
| DMCG-Macro (Zhu et al., 2022) | **88.0** | **0.496** | 40.6 | 0.889 | 55.6 | 0.636 |
| TorDiff-Macro (Jing et al., 2022) | 38.5 | 0.797 | 4.5 | 1.338 | 8.1 | 0.999 |
| **RINGER** | 64.5 | 0.632 | **90.3** | **0.404** | **72.2** | **0.493** |
| | **rRMSD – Recall** | | **rRMSD – Precision** | | **RMSD - F1** | |
| 1-NN (Seq. Sim.) | 35.5 | 0.182 | 20.6 | 0.244 | 26.1 | 0.208 |
| RDKit (Wang et al., 2020) | 8.9 | 0.160 | 0.9 | 0.504 | 1.6 | 0.243 |
| OMEGA (OpenEye, 2022) | 7.1 | 0.163 | 1.3 | 0.525 | 2.2 | 0.249 |
| GeoDiff-Macro (Xu et al., 2022) | 54.2 | 0.120 | 3.0 | 0.559 | 5.7 | 0.198 |
| DMCG-Macro (Zhu et al., 2022) | **89.1** | **0.061** | 32.7 | 0.260 | 47.8 | 0.099 |
| TorDiff-Macro (Jing et al., 2022) | 8.9 | 0.160 | 0.9 | 0.504 | 1.6 | 0.243 |
| **RINGER** | 82.6 | 0.077 | **96.6** | **0.037** | **89.0** | **0.050** |
| | **rTFD – Recall** | | **rTFD – Precision** | | **RMSD - F1** | |
| 1-NN (Seq. Sim.) | 67.7 | 0.054 | 51.1 | 0.078 | 58.2 | 0.064 |
| RDKit (Wang et al., 2020) | 55.3 | 0.051 | 4.4 | 0.206 | 8.2 | 0.082 |
| OMEGA (OpenEye, 2022) | 47.6 | 0.055 | 4.2 | 0.219 | 7.7 | 0.088 |
| GeoDiff-Macro (Xu et al., 2022) | 83.0 | 0.037 | 6.1 | 0.241 | 11.4 | 0.064 |
| DMCG-Macro (Zhu et al., 2022) | **98.2** | **0.017** | 50.6 | 0.097 | 66.8 | 0.029 |
| TorDiff-Macro (Jing et al., 2022) | 55.3 | 0.051 | 4.4 | 0.206 | 8.2 | 0.082 |
| **RINGER** | 90.2 | 0.024 | **98.4** | **0.011** | **94.1** | **0.015** |

## N.4   CYCLIC POSITIONAL ENCODING ABLATION STUDY

To assess the impact of the cyclic relative positional encoding in equation 1 and equation 2, we trained two models on 10% of the training data for 100 epochs: one with a standard relative positional encoding and one with our cyclic relative positional encoding. The results in Tables 9 and 10 illustrate how our newly designed encoding improves performance, especially for larger macrocycles.

Table 9: Ablation study comparing RINGER performance with and without cyclic relative positional encoding defined in equation 1 and equation 2. Models trained on 10% of data for 100 epochs.

| | RMSD – Recall | | | | RMSD – Precision | | | |
|---|---|---|---|---|---|---|---|---|
| | COV ↑ | | MAT ↓ | | COV ↑ | | MAT ↓ | |
| | Mean | Med. | Mean | Med. | Mean | Med. | Mean | Med. |
| Standard encoding | 33.9 | 26.2 | 0.913 | 0.872 | 10.1 | 3.8 | 1.610 | 1.560 |
| Cyclic encoding | **40.4** | **38.6** | **0.861** | **0.819** | **14.2** | **7.4** | **1.538** | **1.491** |
| | **rRMSD – Recall** | | | | **rRMSD – Precision** | | | |
| Standard encoding | 26.2 | 1.387 | 0.213 | 0.191 | 5.8 | 0.1 | 0.721 | 0.701 |
| Cyclic encoding | **32.3** | **13.7** | **0.189** | **0.166** | **9.6** | **0.8** | **0.678** | **0.658** |
| | **rTFD – Recall** | | | | **rTFD – Precision** | | | |
| Standard encoding | 47.2 | 50.0 | 0.064 | 0.055 | 11.0 | 5.1 | 0.168 | 0.149 |
| Cyclic encoding | **55.6** | **62.9** | **0.056** | **0.048** | **16.7** | **10.9** | **0.154** | **0.134** |

Table 10: Ablation study comparing RINGER rTFD performance with and without cyclic relative positional encoding across different macrocycle sizes. Models trained on 10% of data for 100 epochs.

| | | rTFD – Recall | | | | rTFD – Precision | | | |
| | | COV ↑ | | MAT ↓ | | COV ↑ | | MAT ↓ | |
| | #residues | Mean | Med. | Mean | Med. | Mean | Med. | Mean | Med. |
|---|---|---|---|---|---|---|---|---|---|
| Standard encoding | 4 | 62.2 | 76.5 | 0.056 | 0.038 | 18.0 | 20.1 | 0.175 | 0.143 |
| Cyclic encoding | 4 | **69.1** | **84.0** | **0.047** | **0.032** | **25.3** | **29.6** | **0.160** | **0.128** |
| Standard encoding | 5 | 37.1 | 35.6 | 0.066 | 0.062 | 4.1 | 3.0 | 0.151 | 0.141 |
| Cyclic encoding | 5 | **47.0** | **51.4** | **0.061** | **0.055** | **9.0** | **7.5** | **0.137** | **0.127** |
| Standard encoding | 6 | 9.8 | 1.2 | 0.090 | 0.080 | 0.3 | 0.1 | 0.186 | 0.178 |
| Cyclic encoding | 6 | **20.4** | **5.1** | **0.079** | **0.071** | **0.8** | **0.4** | **0.177** | **0.169** |

## O  POST HOC OPTIMIZATION WITH GFN2-XTB

To further evaluate the sampling performance as well as macrocycle conformer quality we performed post hoc optimization with GFN2-xTB. This provides a level comparison between sampling methods. As shown in Tables 11 and 12 below, RINGER maintains excellent performance across recall, precision, and F1 metrics across both all-atom and backbone-only evaluations.

Table 11: *Mean* performance metrics for samples *with post hoc optimization using GFN2-xTB*. Coverage is evaluated at a threshold of 0.75 Å for all-atom RMSD, 0.1 Å for ring-only RMSD (rRMSD), and 0.05 for ring-only TFD (rTFD). *All test data* conformers are used for evaluation.

| | RMSD – Recall | | RMSD – Precision | | RMSD - F1 | |
| Method | COV (%) ↑ | MAT ↓ | COV (%) ↑ | MAT ↓ | COV (%) ↑ | MAT ↓ |
|---|---|---|---|---|---|---|
| RDKit (xTB) | 53.9 | 0.734 | 10.3 | 1.365 | 17.2 | 0.955 |
| OMEGA (xTB) | 44.9 | 0.818 | 8.7 | 1.392 | 14.5 | 1.031 |
| GeoDiff-Macro (xTB) | 29.9 | 0.938 | 3.1 | 1.702 | 5.7 | 1.209 |
| DMCG-Macro (xTB) | **84.9** | **0.415** | 47.4 | 0.856 | 60.8 | 0.559 |
| TorDiff-Macro (xTB) | 62.5 | 0.641 | 12.5 | 1.327 | 20.9 | 0.865 |
| **RINGER (xTB)** | 63.9 | 0.650 | **76.2** | **0.484** | **69.5** | **0.555** |
| | rRMSD – Recall | | rRMSD – Precision | | RMSD - F1 | |
| 1-NN (Seq. Sim.) | 43.7 | 0.301 | 40.3 | 0.331 | 41.9 | 0.315 |
| RDKit (xTB) | 73.0 | 0.098 | 13.4 | 0.508 | 22.7 | 0.164 |
| OMEGA (xTB) | 68.3 | 0.102 | 10.5 | 0.534 | 18.1 | 0.171 |
| GeoDiff-Macro (xTB) | 64.0 | 0.119 | 8.9 | 0.573 | 15.5 | 0.197 |
| DMCG-Macro (xTB) | **93.2** | **0.039** | 53.2 | 0.270 | 67.8 | **0.068** |
| TorDiff-Macro (xTB) | 81.4 | 0.075 | 16.8 | 0.495 | 27.9 | 0.130 |
| **RINGER (xTB)** | 78.7 | 0.101 | **78.9** | **0.119** | **78.8** | 0.109 |
| | rTFD – Recall | | rTFD – Precision | | RMSD - F1 | |
| 1-NN (Seq. Sim.) | 53.1 | 0.111 | 48.6 | 0.122 | 50.7 | 0.116 |
| RDKit (xTB) | 85.6 | 0.029 | 17.3 | 0.200 | 28.8 | 0.051 |
| OMEGA (xTB) | 83.9 | 0.029 | 13.4 | 0.212 | 23.1 | 0.051 |
| GeoDiff-Macro (xTB) | 79.8 | 0.037 | 11.7 | 0.242 | 20.4 | 0.065 |
| DMCG-Macro (xTB) | **96.9** | **0.012** | 57.5 | 0.099 | 72.2 | **0.021** |
| TorDiff-Macro (xTB) | 89.8 | 0.023 | 20.0 | 0.195 | 32.8 | 0.041 |
| **RINGER (xTB)** | 84.1 | 0.032 | **81.7** | **0.042** | **82.9** | 0.037 |

Table 12: *Median* performance metrics for samples *with post hoc optimization using GFN2-xTB*. Coverage is evaluated at a threshold of $0.75\,\text{Å}$ for all-atom RMSD, $0.1\,\text{Å}$ for ring-only RMSD (rRMSD), and $0.05$ for ring-only TFD (rTFD). *All test data* conformers are used for evaluation.

| Method | RMSD – Recall | | RMSD – Precision | | RMSD - F1 | |
|---|---|---|---|---|---|---|
| | COV (%) ↑ | MAT ↓ | COV (%) ↑ | MAT ↓ | COV (%) ↑ | MAT ↓ |
| RDKit (xTB) | 60.1 | 0.688 | 7.0 | 1.311 | 12.6 | 0.903 |
| OMEGA (xTB) | 45.0 | 0.769 | 7.1 | 1.317 | 12.2 | 0.971 |
| GeoDiff-Macro (xTB) | 21.4 | 0.896 | 1.5 | 1.652 | 2.9 | 1.162 |
| DMCG-Macro (xTB) | **92.2** | **0.371** | 50.0 | 0.788 | 64.8 | 0.505 |
| TorDiff-Macro (xTB) | 76.9 | 0.570 | 9.7 | 1.267 | 17.2 | 0.786 |
| **RINGER (xTB)** | 67.6 | 0.590 | **90.2** | **0.361** | **77.3** | **0.448** |
| | rRMSD – Recall | | rRMSD – Precision | | RMSD - F1 | |
| | COV (%) ↑ | MAT ↓ | COV (%) ↑ | MAT ↓ | COV (%) ↑ | MAT ↓ |
| 1-NN (Seq. Sim.) | 35.5 | 0.182 | 20.6 | 0.244 | 26.1 | 0.208 |
| RDKit (xTB) | 91.2 | 0.057 | 9.8 | 0.466 | 17.6 | 0.102 |
| OMEGA (xTB) | 82.5 | 0.070 | 9.0 | 0.501 | 16.2 | 0.123 |
| GeoDiff-Macro (xTB) | 75.1 | 0.086 | 5.6 | 0.541 | 10.4 | 0.148 |
| DMCG-Macro (xTB) | **97.6** | **0.029** | 57.2 | 0.220 | 72.1 | **0.051** |
| TorDiff-Macro (xTB) | 95.9 | 0.041 | 14.3 | 0.454 | 25.0 | 0.075 |
| **RINGER (xTB)** | 85.6 | 0.065 | **91.9** | **0.050** | **88.6** | 0.056 |
| | rTFD – Recall | | rTFD – Precision | | RMSD - F1 | |
| | COV (%) ↑ | MAT ↓ | COV (%) ↑ | MAT ↓ | COV (%) ↑ | MAT ↓ |
| 1-NN (Seq. Sim.) | 67.7 | 0.054 | 51.1 | 0.078 | 58.2 | 0.064 |
| RDKit (xTB) | 98.3 | 0.017 | 14.5 | 0.189 | 25.3 | 0.031 |
| OMEGA (xTB) | 95.8 | 0.021 | 11.7 | 0.204 | 20.8 | 0.038 |
| GeoDiff-Macro (xTB) | 94.0 | 0.026 | 9.3 | 0.233 | 17.0 | 0.047 |
| DMCG-Macro (xTB) | **99.6** | **0.008** | 62.1 | 0.085 | 76.5 | **0.015** |
| TorDiff-Macro (xTB) | 99.4 | 0.012 | 17.8 | 0.184 | 30.2 | 0.023 |
| **RINGER (xTB)** | 90.6 | 0.020 | **94.1** | **0.014** | **92.4** | 0.017 |

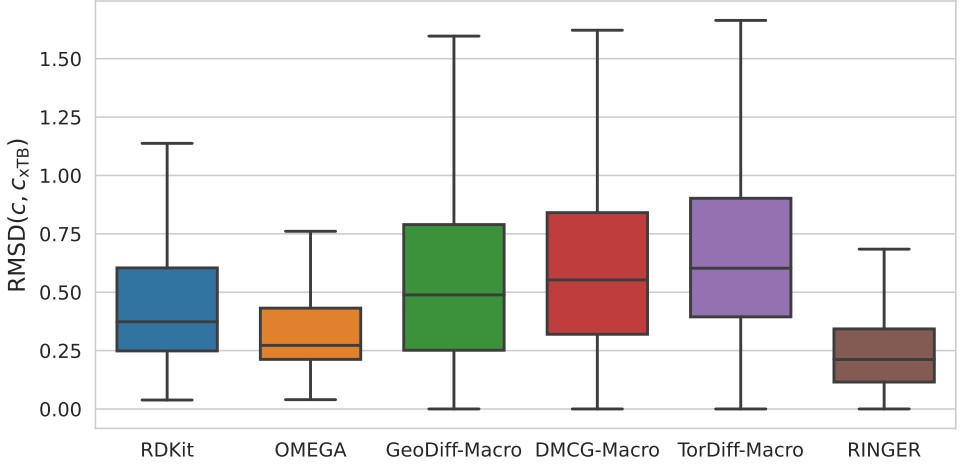

Figure 13: RMSD of generated conformers before and after xTB optimization. On average, RINGER-generated samples require less structural modification to reach xTB local minima. Outliers are not shown for clarity.