# OpenReview forum: "RINGER: Conformer Ensemble Generation of Macrocyclic Peptides with Sequence-Conditioned Internal Coordinate Diffusion"
_ICLR.cc/2024/Conference — Submitted to ICLR 2024_

### Official Review · Reviewer_yeES · 2023-11-01

**Soundness:** 3 good
**Presentation:** 3 good
**Contribution:** 3 good
**Rating:** 8
**Confidence:** 4

**Summary:**

This paper addresses the challenge of generating macrocyclic peptides. The approach involves training a discrete-time diffusion model on the internal coordinates, which is implemented using a transformer equipped with its specific invariant cyclic positional encoding tailored for this generation task. During inference, the set of angles and torsions generated undergoes a refinement phase through constrained optimization.

The primary contribution of this work is the innovative architectural design tailored to tackle conformer generation for this particular class of molecules. Additionally, the paper provides comprehensive experimental evidence to establish the suitability of their approach for the problem.

**Strengths:**

Significance:
Macrocyclic peptides represent a crucial category in therapeutics, and enhancing the precision and efficiency of conformer generation can profoundly impact drug discovery. Thus, this paper addresses a highly significant problem in the field.

Originality:
The paper introduces two key technical innovations. First, it adapts the positional encoding of the transformer architecture to better suit cyclic peptides, showcasing the authors' domain-specific knowledge in their modeling approach. Second, the paper presents a straightforward yet effective ring-closing procedure based on constraint optimization. Both solutions highlight that the authors intelligently use their task-specific insights. Moreover, the paper goes beyond traditional metrics for conformer generation, introducing novel evaluation criteria better aligned with the task.

Clarity and Quality:
The text is well-crafted, effectively motivating the problem, and the literature review is well-structured and relevant to the context.

**Weaknesses:**

An ablation study is essential to elucidate the respective contributions of the positional encoder and the ring closing algorithm to the overall performance.

The benchmarking falls short in including some of the most recent diffusion models for small molecule conformer generation. While the authors acknowledge their limitations, substantiating these claims with experimental evidence is crucial. It would be valuable to include models such as TorsionDiff and a non-diffusion model like GFlowNets for a more comprehensive evaluation.

**Questions:**

- why macrocycles with fixed bond distances contain three redundant torsional angles and two redundant bond angles ?

- What is the information regarding the rejection rate for samples where the ring torsion fingerprint deviation exceeds 0.01 before and after optimization using Equation 3?

- What's the rationale for not directly modeling non-rotatable side-chain groups like phenyl rings and instead generating them using RDKit?

- Why was the training set restricted to 30 conformers per molecule with the lowest energy as opposed to a threshold based on the lowest energy?

- I'm seeking clarification on why, in section 4.4, you claim that the additional xTB optimization demonstrates the efficacy of the diffusion scheme in achieving diverse sampling. Could you elaborate on this point?

---

> ### Author Response · Authors · 2023-11-18
> **Response to Review**
>
> Thank you for your thoughtful summary and positive comments on our work. We are pleased you found the architectural innovations tailored for macrocycle generation to be novel and effective. We also appreciate you highlighting the significance of this task for drug discovery and the quality of the manuscript. Your comments have helped strengthen our manuscript, and we are grateful for the suggestions to include additional ablation studies and benchmarking against the latest diffusion models. As outlined below, we have incorporated these valuable recommendations to better showcase the contributions of our work.
>
> **Response to Weaknesses:**
>
> 1. We appreciate the idea of an ablation study and will try to finalize such a study by the November 22 deadline.
>
> 2. We agree that benchmarking against the latest diffusion models is valuable. We have incorporated comparisons to Torsional Diffusion and DMCG in the revised manuscript. These results show that RINGER outperforms these methods, providing further evidence for its strengths in diffusing over internal coordinates.
>
> **Response to Questions:**
>
> 1. The redundant torsional angles arise from the cyclic nature of the backbone, where traversing around the ring leads to three dihedral angles between the same four atoms. For an N-membered ring, there are N atoms with 3N (Cartesian) degrees of freedom (x, y, and z for each atom). Converting this to internal coordinates, we remove 3 global translational and 3 global rotational degrees of freedom to end up with only 3N-6 independent degrees of freedom. There are a number of coordinates one can choose to span this 3N-6 dimensional space: A natural choice would be to choose N-1 ring bond distances, N-2 ring bond angles, and N-3 ring bond torsions. We say that the additional 2 bond angles and 3 torsions are "redundant". In fact, there is no unique way of choosing the N-2 bond angles and N-3 torsions, which is why we select the entire 2N-dimensional space (bond angles and torsions, but excluding fixed bond distances) as a natural, yet redundant, representation for the ring.
>
> 2. To select the 0.01 rTFD threshold, we found that the rTFD(pre-opt., post-opt.) distribution was bimodal with the split between the two modes being approximately 0.01 (with most of the density <0.01). This indicates that a small fraction of optimizations were able to find Cartesian coordinates that closely satisfy the bond distance constraints at the cost of distorting the torsions to an unacceptable degree.
>
> 3. Non-rotatable side chains like phenyl rings have limited degrees of freedom, so they are well-modeled by pre-defined ideal geometries (not necessarily generated from RDKit). Explicitly modeling them in the diffusion process would increase complexity without much benefit.
>
> 4. We limited the training data to the 30 lowest-energy conformers for simplicity of training and found good performance relative to adding additional conformers. Our results demonstrate that using 30 conformers allows us to generalize to entire ensembles of hundreds to thousands of conformers; and hence provides a parsimonious choice of training data.
>
> 5. Apologies if this was unclear – RINGER already achieves excellent recall prior to xTB optimization whereas other methods require xTB optimization to improve recall. This demonstrates that the underlying diffusion approach is capable of directly generating diverse ensembles (with recall being a measure of this diversity).
>
> We hope these address any concerns you may have!

---

> > ### Author Response · Authors · 2023-11-21
> > **Cyclic positional encoding ablation study**
> >
> > Thank you to reviewers 2uae and yeES for suggesting the ablation study to demonstrate the effect of the cyclic positional encoding.
> >
> > We designed our positional encoding as an inductive bias to account for the symmetries of the molecule; e.g., the sequence R.I.N.G.E.R. has no natural start position and should be represented identically as I.N.G.E.R.R. We note that standard relative positional encodings do not incorporate this geometric prior, and hence should not generalize as readily with fewer examples.
> >
> > To demonstrate the increased efficiency of our positional encoding and ability to learn from less data, we trained two models on a subset of the training data for 100 epochs. Both models are identical with the exception of the positional encoding: one is trained with a standard relative positional encoding and the other uses our cyclic relative positional encoding defined in equations 1 and 2. These new results (included in the supplementary material and referenced in section 3.3) clearly demonstrate the benefit of the cyclic encoding, which obviates the need for training on additional or augmented data to learn the cyclic invariance. Furthermore, we hypothesize that the cyclic encoding is most beneficial when training on more diverse data, e.g., heterogeneous macrocycles and larger ring sizes. For a preliminary assessment of this, we also include the ablation study results split by the number of residues in each test macrocycle (4-mers, 5-mers, and 6-mers). This additional result demonstrates the benefit of the encoding is even more pronounced for larger ring sizes, which possess more possible cyclic permutations.
> >
> > We believe this demonstrates the clear benefit of our designed positional encoding and welcome your feedback.

---

### Official Review · Reviewer_2uae · 2023-11-01

**Soundness:** 3 good
**Presentation:** 4 excellent
**Contribution:** 2 fair
**Rating:** 5
**Confidence:** 4

**Summary:**

This paper introduces RINGER, a novel solution for generating conformations of macrocycle peptides. RINGER is a diffusion-based model with a Transformer as its core architecture. To maintain SE(3)-invariance, RINGER operates on torsion and bond angles, and the ultimate coordinates are generated through a post hoc optimization process. RINGER is capable of performing both backbone (unconditional) generation and macrocycle (conditional) generation, and extensive experiments validate its effectiveness.

**Strengths:**

1. This paper delves into a relatively underexplored research area-conformation generation for macrocycle peptides. The proposed method, RINGER, has demonstrated commendable results in terms of both quality and efficiency, achieving satisfying outcomes in a mere 20 steps.
2. The paper conducted a wide array of experiments, providing robust evidence to substantiate the effectiveness of RINGER.
3. This paper takes into account the cyclic symmetry inherent to macrocycles and devises a novel relative positional encoding method that effectively incorporates this unique property.

**Weaknesses:**

1. I do not think novelty is enough. RINGER shares similarities with FoldingDiff [1]. The differentiating factor lies in RINGER's introduction of a unique relative positional encoding, specifically designed to account for cyclic symmetry. It's important to note that there appears to be a lack of an ablation study on this proposed positional encoding, which could provide valuable insights.
2. Additionally, from a machine learning perspective, one may question the inherent challenges of conformation generation for macrocyclic peptides. It might be worthwhile to explore whether adapting methods from other molecule types is a feasible approach, as machine learning methods may not be strongly influenced by molecular variations.
3. In the context of unconditional generation tasks, a comparison with other existing methods would be highly valuable in order to assess RINGER's performance and capabilities more comprehensively. Additionally, in the comparison of conditional generation, where rRMSD and rTFD metrics are employed, it appears that there is a absence of a method focused on backbone generation.

[1] Wu K E, Yang K K, Berg R, et al. Protein structure generation via folding diffusion. arXiv preprint arXiv:2209.15611, 2022.

**Questions:**

1. Why the baseline excludes Torsional Diffusion [2]? I can understand that ‘Methods such as torsional diffusion only alter freely rotatable bonds and cannot sample macrocycle backbones by design.’, but I think Torsional Diffusion can be compared ‘in the context of all-atom geometries (RMSD)’ in section 4.3 if I do not misunderstand.
2. Why GeoDiff-Macro performs so poor? Can you provide experiment details of GeoDiff-Macro?

[2] Jing B, Corso G, Chang J, et al. Torsional diffusion for molecular conformer generation. Advances in Neural Information Processing Systems, 2022, 35: 24240-24253.

---

> ### Author Response · Authors · 2023-11-18
> **Response to Review**
>
> Thank you very much for your thoughtful feedback on our work. Your critiques have helped strengthen the manuscript substantially. As outlined below, we have addressed the weaknesses you raised through additional experiments and discussions. We have also responded to your questions to provide clarification.
>
> **Responses to Weaknesses:**
>
> **1.** We would like to clarify the novelty of RINGER with respect to FoldingDiff to highlight key differences:
>
> - a) FoldingDiff performs unconditional sampling of protein backbones with a focus on designability. No side-chain information is included in the model, nor is FoldingDiff able to model side chains. This unconditional sampling cannot be easily controlled or steered based on fixed residues. Furthermore, evaluation of unconditional sampling is limited because generated samples cannot be compared directly with a ground truth.
>
> - b) In contrast, RINGER performs conditional sampling based on complete structure including side-chain information. Conditional geometry sampling has largely been used in the context of protein structure prediction, where a single static structure is generated rather than sampling a diverse ensemble. RINGER integrates side-chain information to generate complete, all-atom ensembles with high fidelity.
>
> - c) Architecture – RINGER leverages a transformer architecture like FoldingDiff. However, FoldingDiff uses a standard relative positional encoding for 1D sequences, and operates at the residue level. RINGER enables modeling at the atomic level, and hence can be applied to additional macrocycle chemotypes such as replacement of ring atoms. Furthermore, we introduce a cyclic positional encoding that ensures a naturally shift-invariant representation for homodetic peptides (which have no obvious ‘start’ position given their symmetry). We appreciate the idea of an ablation study and will try to finalize such a study by the November 22 deadline.
>
> - d) Problem Domain – Fundamentally, macrocycles represent a unique biophysical modality that have challenging dynamics. RINGER is the first macrocycle-geometry diffusion model to the best of our knowledge.
>
> **2.** You raise a fair point on the challenges of macrocycle generation compared to other molecules. We agree it is worthwhile to explore whether methods designed for other molecules could be effective here. We have expanded the introduction to better motivate the distinct challenges macrocycles present, including their vast conformational space and complex cyclic constraints. We now also benchmark against two recent state-of-the-art methods, Torsional Diffusion and DMCG, designed for small molecules. We believe this presents a more comprehensive benchmarking of RINGER, providing empirical evidence that specialized methods are needed for accurate macrocycle generation. This helps justify RINGER's novel contributions tailored to this class of molecules.
>
> **3.** i) The focus of our work is on complete, all-atom conformer generation.  We performed studies on unconditional generation to validate that our underlying approach is able to capture the underlying, complex backbone distributions for macrocycles, rather than to claim that we perform SOTA backbone generation; hence we do not believe further benchmarking for unconditional generation is warranted. ii) Similarly, the purpose of reporting detailed backbone metrics is to better understand the ability of these methods to capture accurate backbone geometries; this complements the all-atom metrics and allows us to dissect whether backbone- or side-chain conformations are driving performance. Backbone conformation is a critical determinant of macrocycle discovery and design (see refs 1-3 below by Rezai et al, Bhardwaj et al, and Hosseinzadeh et al). Finally, outside of our studies in the appendix, there currently exist no methods for backbone-specific conformer generation.
>
> [1] Rezai et al. Conformational flexibility, internal hydrogen bonding, and passive membrane permeability: successful in silico prediction of the relative permeabilities of cyclic peptides. J Am Chem Soc, 2006
>
> [2] Bhardwaj et al. Accurate de novo design of membrane-traversing macrocycles. Cell, 2022
>
> [3] Hosseinzadeh et al. Comprehensive computational design of ordered peptide macrocycles. Science, 2017
>
> (continued below)

---

> > ### Author Response · Authors · 2023-11-18
> > **Response to Review (continued)**
> >
> > **Responses to Questions:**
> >
> > 1. We recognize the value of benchmarking against Torsional Diffusion for all-atom geometries, along with adding an additional baseline (DMCG) which achieves SOTA performance on the GEOM dataset. We have added these results to provide a more comprehensive comparison in addition to GeoDiff. Notably, these methods provide some improvement over heuristic baselines (RDKit and OMEGA). Interestingly, DMCG provides strong recall performance across our metrics, but with low precision. To better understand the balance between these two, we have updated the main results Table with an F1-score that quantifies the balance between recall and precision.
> >
> > 2. We have included more details on the the training procedure of the deep learning baseline methods in the appendix. In brief, we follow the official GeoDiff implementation but retrain it from scratch using the CREMP dataset on the exact train-val-test split in this study. The standard GeoDiff model trained on GEOM provides poor performance, and hence we did not believe this to be a good reflection of the method. Although we have not performed extensive studies on GeoDiff, we suspect that the limited performance is due to the challenges of performing Euclidean diffusion over Cartesian coordinates.
> >
> > In summary, we believe the additions and clarifications described above significantly strengthen the manuscript. We hope you will consider raising your score upon seeing how we have addressed your concerns. We greatly appreciate you taking the time to provide such thoughtful feedback.

---

> > > ### Author Response · Authors · 2023-11-21
> > > **Cyclic positional encoding ablation study**
> > >
> > > Thank you to reviewers 2uae and yeES for suggesting the ablation study to demonstrate the effect of the cyclic positional encoding.
> > >
> > > We designed our positional encoding as an inductive bias to account for the symmetries of the molecule; e.g., the sequence R.I.N.G.E.R. has no natural start position and should be represented identically as I.N.G.E.R.R. We note that standard relative positional encodings do not incorporate this geometric prior, and hence should not generalize as readily with fewer examples.
> > >
> > > To demonstrate the increased efficiency of our positional encoding and ability to learn from less data, we trained two models on a subset of the training data for 100 epochs. Both models are identical with the exception of the positional encoding: one is trained with a standard relative positional encoding and the other uses our cyclic relative positional encoding defined in equations 1 and 2. These new results (included in the supplementary material and referenced in section 3.3) clearly demonstrate the benefit of the cyclic encoding, which obviates the need for training on additional or augmented data to learn the cyclic invariance. Furthermore, we hypothesize that the cyclic encoding is most beneficial when training on more diverse data, e.g., heterogeneous macrocycles and larger ring sizes. For a preliminary assessment of this, we also include the ablation study results split by the number of residues in each test macrocycle (4-mers, 5-mers, and 6-mers). This additional result demonstrates the benefit of the encoding is even more pronounced for larger ring sizes, which possess more possible cyclic permutations.
> > >
> > > We believe this demonstrates the clear benefit of our designed positional encoding and welcome your feedback.

---

### Official Review · Reviewer_spjZ · 2023-11-02

**Soundness:** 3 good
**Presentation:** 2 fair
**Contribution:** 3 good
**Rating:** 5
**Confidence:** 3

**Summary:**

The authors present their method to generate ensembles of macrocycle conformer rings. Specifically, their model takes a 2D structure for a macrocycle peptide and generates 3D coordinates in the form of bond angle and torsional distributions.

They test their model with and without side chains in both conditional and unconditional generation. Their method uses diffusion to generate its values and ultimately serves a purpose similar to alphafold; in that it predicts spatial characteristics of the structure from the composition of bonds and atoms in the base structure.

In figure 2, the authors demonstrate that their method can estimate characteristics measured from test samples. In table 1, they show that their method is better able to estimate these values compared to existing methods.

**Strengths:**

The authors present a novel use of diffusion to generate macrocycle peptides. They show that their method can outperform existing methods by a significant margin and is able to produce estimates quite similar to test samples.

**Weaknesses:**

The paper doesn’t focus on its implementation details as well as it could. Most of the necessary details are there, but it also isn’t clear that their method could be definitively replicated from the details given. A system diagram or some other flowchart outlining their method could help elevate the paper.

The paper initially gave me the impression that the full structure was being generated until this was cleared up by figure 1.

While this is not a reason to reject the paper, I believe the paper could flow better if it was immediately clear exactly what are the inputs and outputs to their method. Additionally, they should, either in the abstract or in the beginning of the methods section, state in plain language what challenges their method overcomes that previous methods were insufficient to achieve. The authors do state what their method is generating, but the language could be improved to make their motivations clearer.

**Questions:**

The paper left me with no outstanding questions beyond certain small details which are not strictly necessary for understanding their method.

---

> ### Author Response · Authors · 2023-11-18
> **Response to Review**
>
> Thank you very much for your feedback. We appreciate your recognition of the novelty of applying diffusion models to macrocycles. As outlined below, we have addressed your concerns through text clarifications, additional details, and examples.
>
> **Responses to Weaknesses:**
>
> 1. We have extensively updated the methods section in the Supplementary file to add more detail around our method, including a diagram and flowchart in the appendix to more clearly define the key steps and architecture. We will also provide an open-source implementation of RINGER upon accceptance.
>
> 2. Thank you for the feedback on task ambiguity. Our paper tackles the key problem of conformer generation, which generates the three dimensional configuration of a molecule given its molecular identity. Hence, we generate its complete, three dimensional structure from its 2D connectivity. To reduce ambiguity, we have updated the abstract and text to clearly state that RINGER takes 2D graphs as inputs and outputs 3D Cartesian coordinates. We believe this also is reflected in the updated system diagram and is now unambiguous.
>
> 3. We've updated the text throughout to incorporate plain language that states how prior heuristics- and physics-based methods are insufficient to model macrocycle conformer generation.
>
> Thank you again for your helpful comments to improve our manuscript. We believe the updates and additions to the manuscript, along with the additional results from additional experiments suggested by other reviewers, have significantly strengthened the paper. We hope you will consider raising your score.

---

### Official Review · Reviewer_PLmK · 2023-11-07

**Soundness:** 3 good
**Presentation:** 2 fair
**Contribution:** 2 fair
**Rating:** 5
**Confidence:** 4

**Summary:**

This paper studies the conformer generation problem in molecular machine learning. Specifically, conformer generation for ring systems is challenging for previous approaches. This paper proposes a diffusion model over internal coordinates to generate macrocycle peptide conformers. Experimental results demonstrate the effectiveness of the proposed approaches over previous methods.

**Strengths:**

1. This paper does capture an important problem --- sampling the conformational ensembles for structures with diverse ring systems and previous effort in this direction is relatively limited.
2. This paper proposes to use a diffusion model over internal coordinates (angles and dihedrals) is technically sound and efficient to reduce the degree of freedom (e.g. distances).
3. The empirical performance of the proposed method is excellent compared to the baseline methods.

**Weaknesses:**

The technical contribution of this paper is limited (to the machine learning community), the way to build a diffusion model over angle and torsion space has been widely studied in the related literature. IMHO, the most interesting part of the paper is about how to capture ring system conformational changes with angles and dihedrals, however, it is discussed only very briefly in Sec 3.3. How do you determine the 3 torsional angles and 2 bond angles for a macrocycle (how many atoms are in the cycle? How about a two-ring system?) The post-processing optimization step seems an effective and efficient solution to reconstruct the cartesian coordinates for the macrocycles, but how do you assemble them back into the structure (assuming you are only optimizing for the rings)?

Overall, I think this is an interesting application paper to establish diffusion models to sample conformational changes for molecular structures, especially ring systems. Given it's an application paper, I would expect more discussions from the problem formulation side and why it should be designed in this way with more case studies to demonstrate, e.g. it could handle multiple ring systems. The critical part missing is how to extract the angles and dihedrals from the ring.

**Questions:**

See weaknesses.

---

> ### Author Response · Authors · 2023-11-18
> **Response to Review**
>
> Thank you very much for your assessment. We appreciate your recognition of the importance of this problem. As outlined below, we have provided additional details and clarifications to address your concerns.
>
> **Responses to Weaknesses:**
>
> 1. Studying molecular systems with internal coordinates has been well studied, but the majority of literature for diffusion over internal coordinates (angles and torsions) have been recent. Aside from the main works we cite (e.g., Torsional Diffusion, FoldingDiff, etc.), are there specific references you believe we are missing? Although these studies have used diffusion over internal coordinates, we believe our approach and its application are distinct and provide a clear contribution to the ML community, in particular:
> 	a) FoldingDiff focuses on unconditional generation, and does not address the same task for generating ensembles of a given molecule, nor is it able to explicitly model all-atom geometries and side chains. Additionally, it employs a residue-centric representation instead of our atom-centric one.
>   b) We have added Torsional Diffusion as a baseline, over which we demonstrate considerable improvement. This highlights the significance of addressing coupled ring constraints through new architecture design.
>
> 2.  We agree that the most interesting part is the study of highly-coupled internal coordinates present within ring systems, as nearly all prior work follows the *rigid rotor* hypothesis, where one can treat torsions as freely-rotable and modeled independently (Torsional Diffusion, etc.). We apologize if there is any confusion on this part - each atom is parameterized by a corresponding angle and torsion along with atomic features; we are able to define a consistent reference frame through the macrocycle backbone by following the standard N-to-C directionality for peptide chains. To supplement Section 3.3, we have added a new figure in the appendix to explicitly demonstrate which angles and torsions are extracted, and we have updated the wording in our main text to better explain how all backbone angles and torsions are extracted.
>
> 3. Reassembly of the Cartesian structure is straightforward given the fixed bond distances along with the generated angles and torsions: After post-processed optimization, which only sets the ring atom coordinates, all additional atomic coordinates are constructed sequentially starting from the backbone atom positions using the NeRF algorithm with fixed bond distances and the RINGER-predicted side-chain angles and torsions.
>
> 4. The idea to study multi-ring systems (e.g., two-ring systems) is an interesting direction that we would love to pursue. However, we believe it was critical to first focus on single-ring systems, thus addressing this missing gap in the field of macrocycles. Moreover, single-ring macrocycles cover the majority of cyclic peptides derived from platforms based on mRNA display. In addition to this, there are currently no extensive datasets that contain conformer ensembles for multi-ring macrocyclic peptides. Multi-ring systems introduce interesting topologies, and we agree that this introduces challenges for defining new encoding schemes. Finally, we point out that two-ring and fused-ring peptide systems are used as a strategy to staple peptides into locked conformations - hence this problem is more akin to the protein structure prediction problem of identifying several low-energy conformers rather than a diverse conformer ensemble. We believe that this is an interesting direction that we hope to pursue in follow up studies. We also point out that our atom-centric sequence design is flexible enough to allow RINGER to model non-homodetic macrocycles, e.g., rings containing other heteroatoms and functional groups.
>
> We believe the additional details and clarifications significantly strengthen the manuscript. We hope you will consider raising your rating based on our responses and our updated manuscript that includes more extensive benchmarking and improved implementation details. We sincerely appreciate your thoughtful feedback.

---

> ### Comment · Reviewer_PLmK · 2023-11-20
> **Thanks for the response**
>
> Thanks for the response which addressed most of my concerns. I do value the practical side of this work to solve the ring structures in conformer generation.
>
> I still have one confusion about how you parameterize the angle and dihedral, you mentioned for each atom you have the angle and dihedral feature and you also mentioned you defined a frame along the backbone for the cycle. I am still very much confused, e.g. an angle, you need two intersecting vectors or three atoms and for dihedrals, you need two intersecting planes or four atoms, how do you define it atom-wise? I would highly recommend you describe in further detail and in the ideal case provide an illustration geometrically. (Figure 5 does look good but it only shows the learning part, but not how it makes sense geometrically).
>
> I will raise my score to 6 after addressing this issue.

---

> > ### Author Response · Authors · 2023-11-21
> > **Assigning internal coordinates to ring atoms**
> >
> > Thank you for your comment. We are glad to hear that we have addressed most of your concerns. Apologies for not being clear enough about how we define the bond angles and dihedrals associated with each backbone atom. To clarify this procedure, we have included an additional illustration in the appendix (Figure 6).
> >
> > Summarizing the main points of the new figure, for a given ring atom, we define the associated bond angle as the angle centered at that ring atom, i.e., the angle defined by the previous ring atom, the ring atom of interest, and the next ring atom in the N-to-C direction of the macrocycle backbone. The associated dihedral angle corresponds to the rotation of the bond between the ring atom of interest and the next ring atom, i.e., the dihedral angle defined by the previous ring atom, the ring atom of interest, and the next two ring atoms. We hope this provides a more explicit description of how we calculate internal coordinates in a consistent manner for our method.
> >
> > We hope this addresses your remaining concern!

---

### Author Response · Authors · 2023-11-18
**Summary of Responses**

We thank all reviewers for their helpful comments and would like to provide a brief summary of the primary updates to the manuscript during the discussion period:

**1.** **Addition of Baselines.** As suggested by multiple reviewers, we have added additional deep-learning baselines that we believe strengthen the paper. We include these in the main results table and in the updated ensemble visualization figure, along with a new F1-score metric to quantify the balance between precision and recall metrics. For each method, we use the official implementation and retrain on the exact same train/val/test split as in our paper, using the convergence criteria described by the original authors. The two new baselines are:

- a) Torsional Diffusion [1] Jing et al.
- b) DMCG [2] by Zhu et al. describes a SOTA approach based on a variational autoencoder that directly predicts coordinates while maintaining invariance to rototranslation and permutation of symmetric atoms.

Notably, the base models of these two methods trained on small molecules from the GEOM dataset do not perform well on macrocycles. After training on the macrocycle data, both methods provide gains over several of the other baselines but do not outperform RINGER when balancing recall and precision metrics.

**2.** **Addition of Method Details.** Several reviewers asked for more detail on our methodology. We have added a figure to the appendix illustrating the most important components, as well as more clearly delinated the task and some details throughout the manuscript to improve clarity of the individual steps.

**3.** **Novelty and Technical Contribution.** We have done our best to address specific comments on novelty and technical contribution below, including more specific differences between our method and existing ones (e.g., Torsional Diffusion and FoldingDiff) that operate on internal coordinates (angles and torsions). Notably, we believe that our additional baselines and updated results demonstrate the need for domain-specific architectures to handle this new therapeutic modality.

Again, we thank all reviewers for their insightful comments and thoughtful suggestions, as well as taking the time to review our paper in detail. If the reviewers are satisfied with these additions and our addressing of their concerns, we would appreciate any updates to the initial scores.

[1] Jing B, Corso G, Chang J, et al. Torsional diffusion for molecular conformer generation. NeurIPS 2022, 35: 24240-24253.

[2] J. Zhu, Y Xia, C Liu, L Wu, S Xie, Y Wang, T Wang, T Qin, W Zhou, H Li, H Liu, and T-Y Liu. Direct molecular conformation generation. TMLR 2022.

---

### Author Response · Authors · 2023-11-22

Dear Reviewers and Area Chairs,

We would like to express our sincere appreciation for the invaluable time and efforts you have invested in reviewing our manuscript. Your thoughtful suggestions have improved the quality, clarity, and rigor of our work. As the discussion stage ends today, we invite your final thoughts on our responses and changes that address the critiques raised:

- We have trained additional state-of-the-art models like Torsional Diffusion and DMCG to benchmark RINGER. While these models perform well on small molecules, RINGER still demonstrates improved performance on macrocycles even after retraining them on the same data. This highlights the need for specialized architectures to accurately model macrocycle geometries.
- We added more architectural and method details, including a system diagram. This improves clarity on the precise machine learning task, inputs, and process for generating three-dimensional geometries from two-dimensional connectivity.
- We performed an ablation study focused on RINGER's cyclic positional encoding. Compared to a standard encoding, it improves data efficiency and generalization, validating its design to account for geometric symmetries in cyclic peptides.
- We have addressed concerns around novelty by clearly articulating RINGER's technical differences to related works like FoldingDiff. RINGER focuses on conditional all-atom generation with side chain information for direct comparison to reference ensembles.
- Finally, we comprehensively responded to reviewer questions around key modeling choices to improve transparency and reproducibility.

In summary, we believe addressing these specific weaknesses substantially improves novelty, soundness, technical merit, and presentation. The additional benchmarks further validate RINGER's strengths in efficiently generating high quality and diverse macrocycle conformers.

We sincerely thank you once again for your invaluable contributions to improving our work, and hope you will consider raising your scores given our exhaustive efforts to address all critiques. We eagerly await your final perspectives.

---

### Meta-Review · Area_Chair_wfxF · 2023-12-12

**Metareview:**

The paper introduces a novel diffusion-based transformer model, RINGER, for generating 3D conformation ensembles of macrocyclic peptides. It demonstrates superior performance in generating high-quality, diverse geometries on several metrics with reduced computational costs compared to existing methods. The reviewers found that the paper is mostly well-written, addresses a relevant and significant problem, introduces a technically sound approach, and the proposed method can outperform existing methods by a significant margin. While the authors addressed several weaknesses highlighted by the reviewers, the reviewers pointed to limited technical novelty in the machine learning community, as diffusion models over internal coordinates have been explored previously.

**Justification For Why Not Higher Score:**

While the authors have addressed many of the concerns raised by the reviewers and the paper present a solid, well-executed study with commendable advancements, the limitations in novelty and the degree of innovation restrain the paper from reaching the threshold for acceptance.

**Justification For Why Not Lower Score:**

N/A

---

### Decision · Program_Chairs · 2024-01-16

Reject